# Ratiometric afterglow luminescent nanoplatform enables reliable quantification and molecular imaging

Yongchao Liu[1], Lili Teng[1], Yifan Lyu[1], Guosheng Song [1✉], Xiao-Bing Zhang [1✉] & Weihong Tan [1]

Afterglow luminescence is an internal luminescence pathway that occurs after photo-excitation, holds great promise for non-background molecular imaging in vivo, but suffer from poor quantitative ability owing to luminescent attenuation over time. Moreover, the inert structure and insufficient reactive sites of current afterglow materials make it hard to design activatable afterglow probes for specific detection. Here, we report a ratiometric afterglow luminescent nanoplatform to customize various activatable afterglow probes for reliable quantification and molecular imaging of specific analytes, such as NO, $ONOO^-$ or pH. Notably, these afterglow probes can not only address the attenuation of afterglow intensity and eliminate the interference of factors (e.g., laser power, irradiation time, and exposure time), but also significantly improve the imaging reliability in vivo and signal-to-background ratios (~1200-fold), both of which enable more reliable quantitative analysis in biological systems. Moreover, as a proof-of-concept, we successfully design an NO-responsive ratiometric afterglow nanoprobe, RAN1. This nanoprobe can monitor the fluctuations of intratumoral NO, as a biomarker of macrophage polarization, making it possible to real-time dynamically evaluate the degree cancer immunotherapy, which provides a reliable parameter to predict the immunotherapeutic effect.

---

[1] Molecular Science and Biomedicine Laboratory (MBL), State Key Laboratory of Chemo/Biosensing and Chemometrics, College of Chemistry and Chemical Engineering, Hunan University, 410082 Changsha, P. R. China. ✉email: songgs@hnu.edu.cn; xbzhang@hnu.edu.cn

Low-background imaging and accurate measurement of specific analytes in biological systems are critical to basic biomedical research and clinical application[1–4]. Fluorescent imaging suffers from inevitable photobleaching and relatively high autofluorescence background of biological samples[5,6]. In contrast, afterglow luminescence is an excitation-free imaging technology that stores irradiated photoenergy and then slowly emits photons[7–11]. As a result, it can eliminate the disadvantages of fluorescent imaging and provide, instead, non-background molecular imaging in vivo[7,12,13]. The significantly improved signal-to-background ratio (SBR) and imaging quality combine to make afterglow imaging a powerful alternative for intravital imaging[13,14], such as cell tracking[15,16], cancer imaging[17,18], lymph node mapping[19], visualization of vascularization[20], monitoring temperature in vivo[21] and drug-induced hepatotoxicity[8], and so on. Nonetheless, two challenges limit the biological applications of afterglow materials. First, because of the structural inertness of current afterglow materials, not enough response sites are available to design activatable afterglow probes toward different biological species or physiological processes[20,22–24]. Second, the attenuation of afterglow intensity during molecular imaging makes it difficult to accurately quantify analytes, such as disease biomarkers, ions, and biological messengers, especially in living systems[7,25,26]. Therefore, providing reliable and specific quantitation of various biotargets in physiological or pathological processes demands a de novo activatable afterglow-based imaging platform. Presently, Förster resonance energy transfer (FRET) is a widely used self-calibration strategy for quantitative detection and analysis, during which one fluorophore (donor) transfers its excited-state energy to another fluorophore (acceptor), and the latter usually emits red-shifted fluorescence[27–29]. Since the donor and acceptor are two molecules with different reactive sites, FRET can allow more flexible probe design by introducing different target-responsive probes as the energy donor or acceptor[30,31].

We hypothesized the development of a novel afterglow imaging nanoplatform that would (1) solve the problems inherent in the use of afterglow materials and (2) allow for the customization of activatable afterglow probes for reliable quantification and molecular imaging of specific analytes. To accomplish this, we herein report a de novo afterglow-based energy transfer system, termed as "afterglow resonance energy transfer" (ARET), which combines the advantages of FRET and afterglow luminescence for a universal afterglow-based sensor design and quantitative imaging. In ARET, the donor fluorophore of FRET is replaced by an excitation-free afterglow substrate so that the resonance energy transfer is between the afterglow substrate (energy donor) and the acceptor fluorophore. Upon irradiation, the afterglow substrate generates an excited-state intermediate (energy donor) to store the energy. Then, the stored energy is transferred to the energy acceptor to emit another afterglow after light cessation. As a result, ratiometric afterglow imaging can be implemented by self-calibration of two (or more) emissions, and the problems caused by afterglow attenuation can be effectively solved. Using ARET strategy, we next designed a universal ratiometric afterglow nanoplatform (RAN) to customize activatable afterglow probes for reliable and specific biotarget quantification, through integrating the responsive molecules (NRM, ORM or PRM), afterglow substrate (MEHPPV), surfactants (F127) and afterglow initiators (TPP or BDP) via a self-assembly strategy (Fig. 1a and Supplementary 2). By introducing different target-responsive molecules as the energy acceptor (Fig. 1b), various activatable ratiometric afterglow nanoprobes could be easily prepared for sensing and imaging of the analytes such as NO, ONOO−, or pH in systems. As a biological proof-of-concept, RAN1, our test case of NO-responsive afterglow nanoprobe for NO imaging, which showed a decreasing afterglow emission at 600 nm and increasing afterglow emission at 830 nm upon activation of NO, was constructed based on the ARET strategy. Such ratiometric strategy could provide reliable and quantitative analyses by self-calibration of the two emissions noted above. Next, using high SBR imaging, we tested the ability of RAN1 to specifically detect NO released from activated M1-phenotype macrophages in the tumor microenvironment (TME) (Fig. 1c). We found that ratiometric signals could, indeed, be used as a noninvasive predictor for real-time evaluation of macrophage-mediated tumor immunotherapy.

## Results

**Construction of ratiometric afterglow nanoplatform.** Poly[2-methoxy-5-(2-ethylhexyloxy)-1,4-phenylenevinylene] (MEHPPV) was selected as the afterglow substrate to release delayed luminescence in the form of photons[8]. Afterglow initiators (AI: BDP and TPP)[32–34] capable of generating $^1O_2$ were used to initiate the afterglow of MEHPPV (energy donor) by forming unstable PPV-dioxetane, which acted as an intermediate to successively transfer the afterglow energy to the responsive molecules (energy acceptor) with longer emitting wavelength through the ARET process. Thus, after light cessation, MEHPPV was initiated by AI to emit afterglow (AF1), and the energy of AF1 was transferred to the target-responsive molecules (energy acceptor) to release afterglow with longer wavelength (AF2) by the $A_{12}RET$ process (Fig. 2a). As a result, the quantitative detection of analytes could be performed by calculating the afterglow intensity ratio between the afterglow donor and acceptor where ARET occurs. The synthetic route was outlined in Supplementary Fig. 1 and the new compound structures were confirmed by MS and NMR (Supporting Note).

First, to realize afterglow sensing of nitric oxide (NO), a NO-responsive molecule (NRM) was selected as the energy acceptor to construct RAN1. Owing to intramolecular charge transfer (ICT), the weak electron acceptor (benzo[c][1,2,5]thiadiazole-5,6-diamine) in NRM was oxidized by NO to generate a stronger acceptor (5H-[1,2,3]triazolo[4,5-f]-2,1,3-benzothiadiazole) in NRM-NO (Fig. 1b). Then, the enhanced effects of ICT resulted in a red-shift of the absorption and emission wavelengths of the generated NRM-NO[35]. According to the sensing mechanism shown in Fig. 2b, RAN1 could generate two corresponding afterglow signals (AF1 from MEHPPV and AF2 from NRM-NO). The quantitative detection of NO could then be achieved by calculating the ratio (AF2/AF1) between the AF1 and AF2.

Dynamic light scattering (DLS) showed the size and Zata potential of RAN1, and the transmission electron microscopy (TEM) image revealed its spherical morphology (Supplementary Fig. 3). After optimizing parameters for syntheses, such as afterglow initiator choice and the doping amount of NRM or surfactant, and measurement condition, such as buffer medium, pH values, and signal acquisition modes, of RAN1 (Supplementary Figs. 4–8), we then systematically studied its response to NO. As expected, with the increased concentrations of NO, the absorbance at 660 nm gradually increased, and the fluorescence of RAN1 at 830 nm increased significantly, while the emission peak at 600 nm decreased drastically (Fig. 2c and Supplementary Fig. 9). After the incubated solutions were pre-irradiated by a 660-nm laser, the afterglow and fluorescent images showed brighter AF2 (830 nm) and darker AF1 (600 nm) with the increase of NO concentration (Fig. 2d and Supplementary Fig. 10). These spectral changes indicated the response of RAN1 to NO and efficient energy transfer from AF1 to AF2. Moreover, the afterglow ratio (AF2/AF1) was linearly correlated with NO concentrations from 0 to 20 μM, and the detection limit (3σ/slope) of RAN1 for NO was calculated to be 0.21 μM (Supplementary Fig. 11), strongly suggesting that RAN1 could be used for NO quantification. Notably, the reaction kinetics

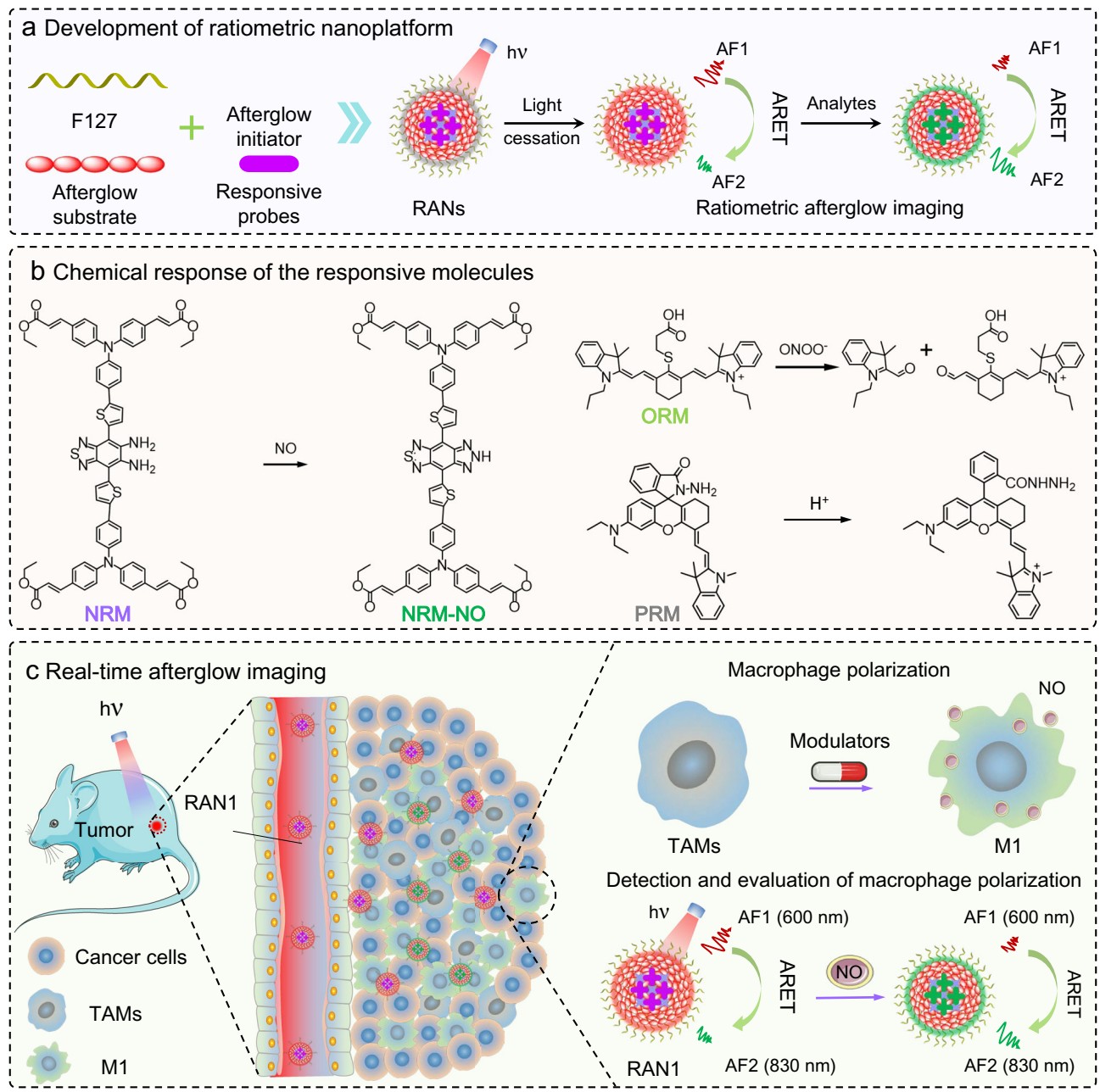

**Fig. 1 Design of ratiometric luminescent nanoplatform for reliable imaging and evaluation of macrophage polarization. a** Schematic illustration of ARET-based ratiometric nanoplatform. **b** Chemical structures of the responsive molecules (NRM, ORM, and PRM) before and after response to NO for NRM, or ONOO⁻ for ORM, or pH for PRM. **c** Real-time afterglow imaging of macrophage polarization.

of RAN1 to NO showed that the absorbance at 660 nm reached a plateau within 30 min, and the good dispersion of RAN1 in DPBS and DMEM culture medium with diameters of ~45 nm during 5 days indicated its good colloid stability (Supplementary Fig. 12). Next, we measured the specificity of RAN1 toward NO and found that only NO could induce the significant enhancement of AF2/AF1 and FL2/FL1 ratios, indicating that RAN1 was a specific probe for the detection of NO (Supplementary Fig. 13). The absorption peak of MEHPPV at 500 nm showed a strong hypochromatic shift, and an intensity decrease was observed after light irradiation, indicating degradation of conjugation structure for MEHPPV and the generation of PPV-dioxetane[10] (Supplementary Fig. 14). These results indicated that RAN1 was an excellent nanoprobe for afterglow imaging of NO with high sensitivity and specificity.

**The universality of the ratiometric activatable afterglow nanoplatform.** Smart stimuli-responsive afterglow materials are promising contrast agents for the development of next-generation molecular probes in imaging, diagnostics, tissue engineering, and biomedical devices[36–38]. Benefiting from ARET, RAN is a universal platform that can be used to customize different activatable ratiometric afterglow probes toward various biotargets. To prove the flexibility of our strategy, we then tried to develop two more stimuli-responsive afterglow probes, RAN2 and RAN3, which were constructed by using another two responsive molecules, ORM (ONOO⁻-responsive molecule) and PRM (pH-responsive molecule)[38,39], respectively. The chemical structures and ratiometric sensing mechanisms are shown in Fig. 2b and Supplementary Fig. 2.

For RAN2, the DLS and TEM image showed the size and spherical morphology of RAN2 in diameter (Supplementary Fig. 15).

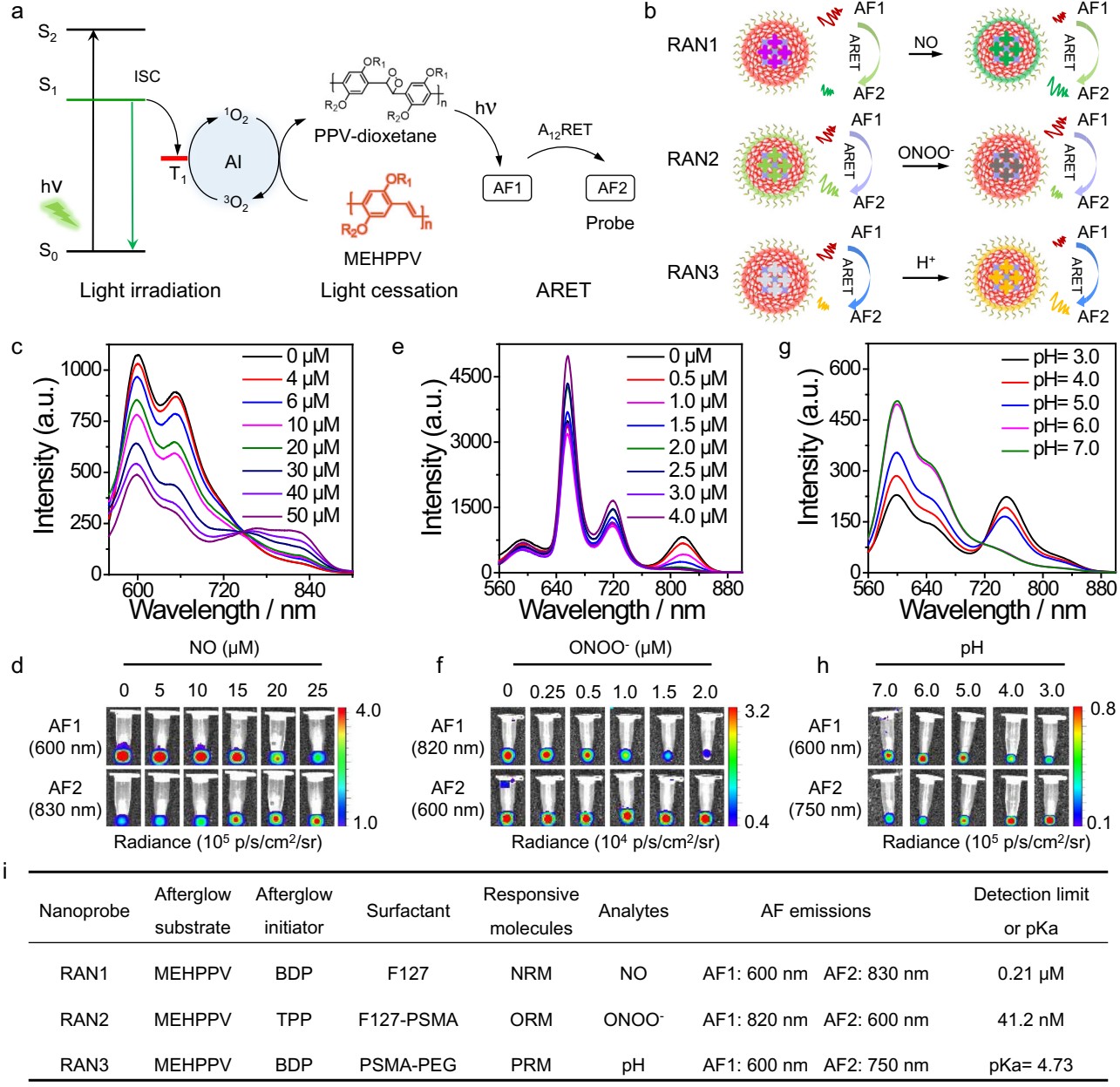

**Fig. 2 Characterization of RANs and their ratiometric response in vitro. a** Schematic diagram of the probable responsive mechanism of afterglow probes. AF1 and AF2 represent the afterglow for afterglow substrate (MEHPPV) and responsive molecules, respectively. $A_{12}$RET represents the afterglow resonance energy transfer from AF1 to AF2. **b** General schematic illustration of RANs for ratiometric afterglow sensing. **c–e** Fluorescent emission spectra of RAN1-3 upon activation of NO, ONOO$^-$, and pH, respectively. **f–h** Afterglow images of RAN1-3 upon activation of NO, ONOO$^-$ and pH, respectively. **i** The constituent units and detection capability of RAN1-3. Source data are provided as a Source Data file.

With the increase of doping amounts of ORM, the fluorescent and afterglow images showed the decreased intensity of AF2 or FL2 and increased intensity of AF1 or FL1, suggesting the effective energy transfer from AF1 to AF2 (Supplementary Fig. 16). With the increase of ONOO$^-$ concentration, the absorption and fluorescent intensity at 700–850 nm both gradually decreased, as well as the afterglow and fluorescent images of ORM showed darker contrast in AF1 (820 nm) and constant in AF2 (600 nm) (Fig. 2e, f and Supplementary Fig. 17), suggesting the ratiometric response of RAN2 to ONOO$^-$ within the nanoparticle. The reaction kinetics of RAN2 to ONOO$^-$ showed the quick response of RAN2 to ONOO$^-$, and the linear relationship confirmed that RAN2 could be used for ONOO$^-$ quantification with a detection limit ($3\sigma$/slope) of 41.2 nM (Supplementary Fig. 18). In addition, the fluorescent

spectra and afterglow images demonstrated the excellent ability of RAN3 to detect pH values through ratiometric imaging (Fig. 2g, h). The spectra changes of absorption and fluorescence emission suggested the good responsiveness of RANs toward NO, ONOO$^-$, and pH, respectively (Supplementary Fig. 19), and the responsive difference between ratiometric afterglow nanoplatform and responsive molecules (NRM, ORM or PRM) demonstrated that the selection of responsive molecules with larger wavelength gap with MEHPPV can reduce the spectral overlap between them, which is conducive to design ratiometric afterglow probes with better responsiveness (Supplementary Fig. 20). These results indicated that the ARET-based strategy was also suitable for the quantitative detection and imaging of ONOO$^-$ or pH with high sensitivity, further confirming the feasibility and universality of the ratiometric

afterglow nanoplatform for developing afterglow probes. Therefore, taking afterglow substrate (MEHPPV) and afterglow initiator (BDP and TPP) as the afterglow generation unit, and introducing responsive molecular probes (NRM, ORM, and PRM) as the detection unit, we could achieve quantitative detection of analytes (NO, ONOO⁻, and pH) based on the afterglow emission ratio (AF2/AF1) of donor and acceptor through our ARET strategy (Fig. 2i). Theoretically, we could also customize various ratiometric afterglow probes by introducing responsive units that match the spectrum of the afterglow substrate.

**The reliability of RAN-based ratiometric afterglow sensing.** Notably, the attenuation of afterglow intensity over time after light cessation makes it difficult to reliably quantify specific analytes[15,23]. Moreover, afterglow luminescent intensity is also dependent on laser power, irradiation time, and exposure time. It was assumed that the ratios of two afterglow luminescent intensities would be independent of those two interference factors, thereby enhancing reliability for accurate quantification (Fig. 3a). To confirm the supposition, we systematically investigated the afterglow luminescent intensity ratio of RAN1 under different laser power (Fig. 3b), irradiation time (Fig. 3c), and acquisition time (Fig. 3d) during NO response. As expected, the afterglow intensities of both AF1 (600 nm) and AF2 (830 nm) were proportional to the laser power and irradiation time, but inversely proportional to the exposure time. Surprisingly, the afterglow intensity ratio of AF2/AF1 was constant and independent of them. Similarly, for RAN2, the AF2/AF1 ratios were also independent of laser power, irradiation time, and acquisition time, and only depend on ONOO⁻ (Supplementary Fig. 21). Then, we explored the decayed afterglow intensity of RAN1 under different NO concentrations (Fig. 3e). With the increase of decay time, the intensity of both AF1 and AF2 showed a similar trend of decay, irrespective of the presence of NO (Fig. 3f and Supplementary Fig. 22). In contrast, the AF2/AF1 ratio displayed no obvious change with the increase of decay time and was dependent on the concentration of NO (Fig. 3g), which indicated the reliability of using the AF2/AF1 ratio to quantify NO concentration, compared with AF1 or AF2 intensity alone.

We then explored the attenuated afterglow intensity of RAN1 at different doping amounts of NRM (Fig. 3h). Notably, even if the doping amounts of NRM are increased, the results confirm that the attenuation of afterglow cannot be prevented. Nonetheless, the attenuation of afterglow still cannot affect the afterglow ratio (AF2/AF1) of RAN1 because it was nearly constant as attenuation time increased and only proportional to the doping amounts of NRM, which offered a feasible strategy to regulate the afterglow ratio (Fig. 3i and Supplementary Fig. 23). As shown in Fig. 3j, the intensity of both AF1 and AF2 was decreased with the increasing attenuation time, while the normalized afterglow ratio (AF2/AF1) was constant. These results confirmed that the intensities of single afterglow luminescence were affected by laser power, irradiation time, acquisition time, or attenuation time. In contrast, the AF2/AF1 ratios were independent of those parameters, only correlating with the concentration of NO. Therefore, afterglow ratio (AF2/AF1) can serve as a reliable indicator for calculating the real level of NO.

Furthermore, we explored the luminescence ability of RAN1 and its response to NO in tissues with different thicknesses. When chicken tissues were placed on top of RAN1, the decreased afterglow (AF1 and AF2) and fluorescent (FL1 and FL2) signals were observed, irrespective of the presence of NO (Fig. 3k, l and Supplementary Fig. 24a, b). Owing to the lack of real-time excitation for afterglow, the SBR of afterglow was significantly higher than that of fluorescence (~1200-fold

for AF1). Specifically, SBR of fluorescence was significantly decreased to the level of background noise for tissue thickness beyond 0.2 cm, whereas SBR for afterglow was still very high (Fig. 3n and Supplementary Fig. 24d). Importantly, the afterglow intensity ratios (AF2/AF1) were constant for any depth from 0 to 0.6 cm, whether in the presence of NO or not, while the fluorescence intensity ratios (FL2/FL1) were decreased with the increasing depth (Fig. 3m, n and Supplementary Fig. 24c). And the main reason for the difference in fluorescence and afterglow ratio may be attributed to the background signal of chicken tissues. These results indicated that the ratiometric afterglow probe has greater reliability when imaging the deep-sited tissues, compared with that using fluorescence or the single emission of afterglow.

**Ratiometric afterglow imaging in vivo.** To examine RAN1 for in vivo afterglow imaging of NO, the inflamed mouse model was established. Briefly, 5 mg/mL of LPS were used to cause the inflamed model in the right rear paws of mice via intramuscular (i.m.) injections[40] (Fig. 4a). As expected, the luminescent signals of inflamed rear paws were gradually enhanced over time (Fig. 4b and Supplementary Fig. 25a). After quantification, both the fluorescent and afterglow intensity ratios in inflammatory areas showed quicker increase than that of PBS-treated areas from 1 to 3 h (Fig. 4c and Supplementary Fig. 25b), indicating that RAN1 could detect endogenous NO within the inflamed region through ratiometric afterglow imaging. Moreover, the afterglow signals presented higher SBR than that of fluorescence (Supplementary Fig. 25c), which was attributed to the reduced background from afterglow images of mice.

To further confirm the capability of RAN1 for imaging NO, Nos2⁻/⁻ mice (those mice with the Nos2 gene knocked out can no longer express nitric oxide synthase 2 (Nos2)) were used as a control in the LPS-induced liver injury experiment. From the afterglow and fluorescence images and the quantification signal from liver areas (Fig. 4d, e and Supplementary Fig. 26), as for WT mice, the treatment of LPS could significantly enhance the afterglow intensity ratio (AF2/AF1), in contrast to PBS treatment, indicating the high content of NO in the liver of LPS-incubated WT mice group. As expected, as for Nos2⁻/⁻ mice, the treatment of LPS induced no notable increase of AF2/AF1, which was consistent with no obvious production of NO in LPS treated Nos2⁻/⁻ mice due to the knocked out of Nos2 gene. Furthermore, the flow data and immunofluorescence staining of intracellular iNOS showed the higher expression of iNOS in LPS-incubated WT mice, while no obvious expression of iNOS in LPS-incubated Nos2⁻/⁻ mice (Fig. 4f and Supplementary Fig. 27). From those comparison experiments, it was concluded that RAN1 was able to detect endogenous NO in living mice via ratiometric afterglow imaging.

The reliability of ratiometric afterglow imaging after light cessation was further investigated in mice (Fig. 4g). Both AF1 and AF2 intensities were reduced with increasing attenuation time, while the normalized AF2/AF1 ratios were constant over time (Fig. 4h), which further confirmed the reliability of such ratiometric afterglow in vivo. Furthermore, we explored the ability of RAN1 to accumulate in tumors through the enhanced permeability and retention (EPR) effect by recording the fluorescence and afterglow intensities at the indicated time points (Fig. 4i). Notably, the normalized intensity ratios for both fluorescence and afterglow exhibited a gradual increase from 1 to 18 h, suggesting that the ratiometric imaging of RAN1 could achieve the detection of endogenous NO within the TME. Moreover, the higher SBR of afterglow than that of fluorescence confirmed its higher imaging reliability (Fig. 4j). Because of the

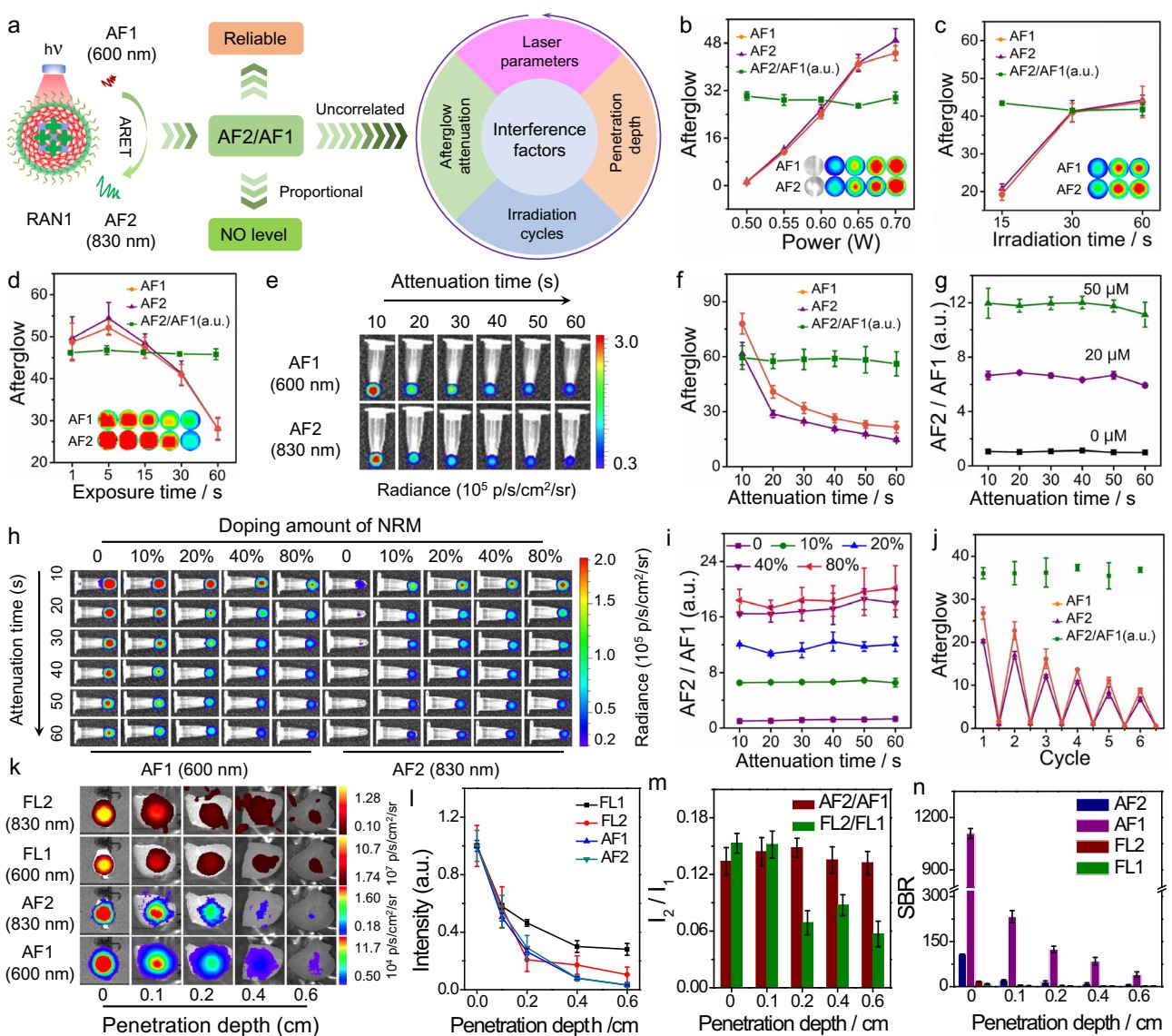

**Fig. 3 The reliability of ratiometric afterglow sensing for RAN1 solution. a** Schematic illustration of RAN1 for reliable afterglow sensing of NO. **b–d** The afterglow intensities (AF1, AF2) and ratio (AF2/AF1) for RAN1 (20 μg/mL) as a function of (**b**) laser power, (**c**) irradiation time, and (**d**) exposure time. The insert is the corresponding afterglow images for AF1 and AF2, respectively. **e** Representative afterglow images of RAN1 (20 μg/mL) as a function of decayed time in the presence of NO (50 μM). **f** Quantification of AF1, AF2 and AF2/AF1 ratio of RAN1 in the presence of NO (**e**). **g** Normalized AF2/AF1 ratio for RAN1 at different NO concentrations. **h** Representative afterglow images of RAN1 (20 μg/mL) with different doping amounts of NRM, as a function of decayed time in the presence of NO (50 μM). **i** Corresponding AF2/AF1 ratios in (**h**). **j** AF1, AF2, and AF2/AF1 ratio from RAN1, as a function of light irradiation cycles. **k** Fluorescent and afterglow images of RAN1 (20 μg/mL) through chicken tissues of different thicknesses in the absence of NO. **l** Normalized fluorescence intensities (FL1 and FL2) and afterglow luminescence intensities (AF1 and AF2), as a function of penetration depth, in the absence of NO. The signal intensity of chicken tissue with a thickness at 0 cm was defined as one unit. **m** Corresponding fluorescence ratio (FL2/FL1) and afterglow ratio (AF2/AF1), as a function of penetration depth. **n** SBR for FL1, FL2, AF1, and AF2, as function of penetration depth, in the absence of NO, respectively. Data are presented as mean values ± s.d. (*n* = 3). Source data are provided as a Source Data file.

lower background signals of afterglow from mice, the afterglow ratios AF2/AF1 showed a fast increase over time (Fig. 4k), compared with fluorescence ratios (FL2/FL1). These results demonstrated the ability of RAN1 for real-time afterglow imaging of endogenous NO variation for tumor-bearing mice. Moreover, RAN2 was also employed for in vivo imaging of acetaminophen (APAP)-induced hepatotoxicity[41]. From the afterglow and fluorescent images (Supplementary Fig. 28), both the afterglow and fluorescent intensity ratios for APAP-treated mice were higher than that of PBS, respectively, which indicated that RAN2 could detect ONOO⁻ in mice in real-time and be used to evaluate modulator-induced hepatotoxicity in vivo.

**Afterglow imaging of macrophage-modulated immunotherapy.** Owing to the excellent properties of RAN1 for reliable imaging of NO in solution, we expected the potential of RAN1 to image the polarization of RAW264.7 macrophages (Fig. 5a). First, the suitability of the afterglow nanoprobe (RAN1) for biological applications was confirmed by the low cytotoxicity (Supplementary Fig. 29). Colocalization experiments in RAW264.7 macrophages showed that RAN1 mainly located in lysosomes (Supplementary Fig. 30). To modulate the polarization of macrophages, four modulators, including interferon-γ (IFN-γ), BLZ945, pexidartinib and chloroquine, were employed to stimulate RAW264.7 macrophages for 24 h[42–45] (Supplementary

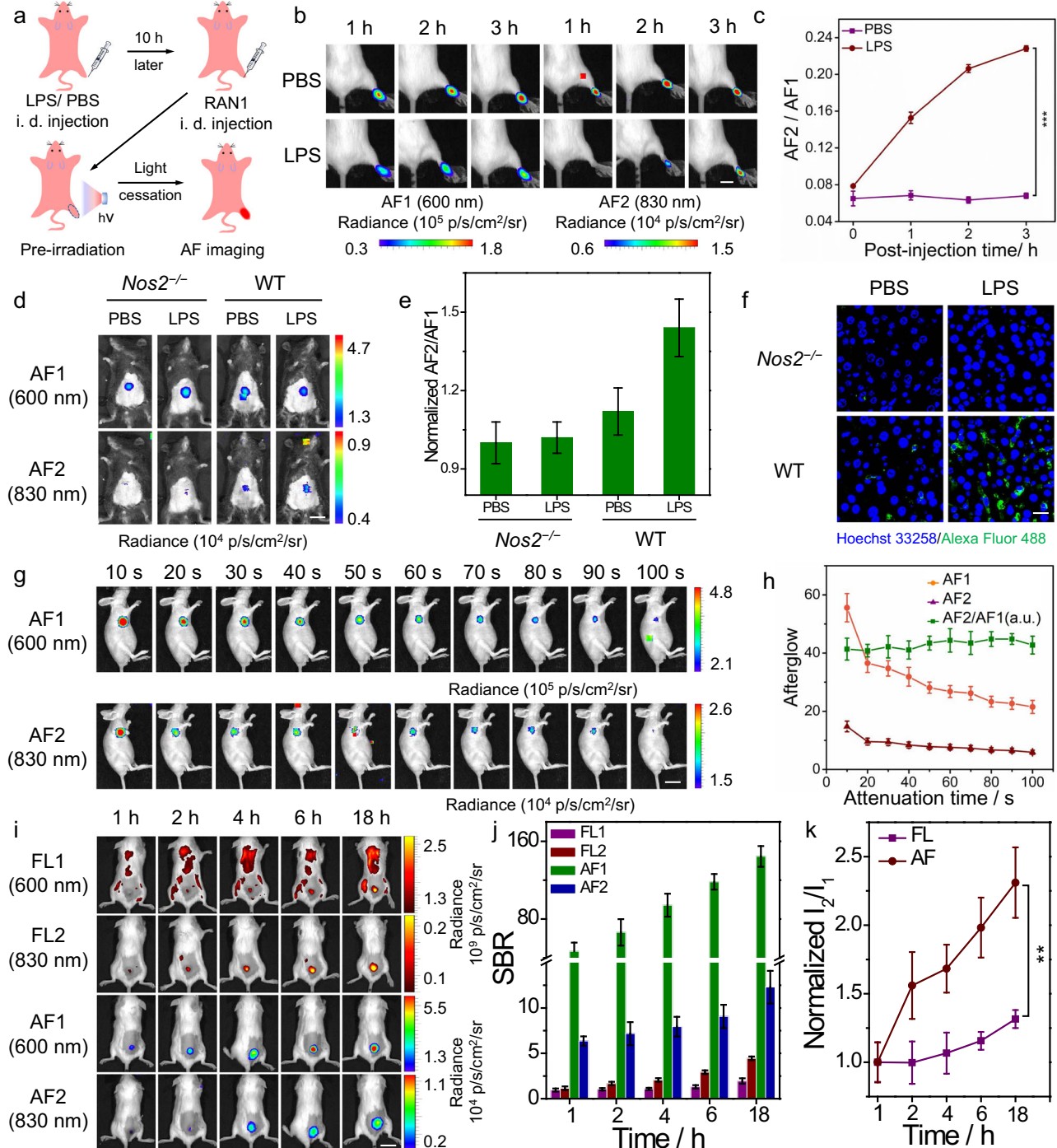

**Fig. 4 Ratiometric afterglow imaging in vivo. a** Schematic illustration showing luminescence imaging of NO in LPS-induced inflammation models. **b** Representative afterglow images, as a function of post-injection time of RAN1 (10 μg/mL) in LPS, or PBS-pretreated mice. Scale bar: 1 cm. **c** The corresponding afterglow intensity ratios from (**b**). Data are presented as mean values ± s.d. (***$P < 0.001$, $n = 3$). Statistical differences were analyzed by Student's $t$ test. **d** Representative afterglow images of $Nos2^{-/-}$ and WT mice upon post-intravenous injection of RAN1 (200 μg/mL) in the LPS-induced liver injury model. Scale bar: 2 cm. **e** Normalized afterglow intensity ratio in (**d**). **f** Representative immunofluorescence staining of iNOS in liver slices of $Nos2^{-/-}$ and WT mice upon different administrations. Green signals indicate iNOS stained with Alexa Fluor 488-labeled iNOS antibody and blue signals represent the cell nucleus. Scale bar: 20 μm. Each experiment was repeated for three times. **g** Representative afterglow images of 4T1 tumor-bearing mice, as a function of decayed time at 6 h post-intravenous injection of RAN1 (200 μg/mL). Scale bar: 2 cm. **h** Quantification of afterglow intensities and normalized intensity ratio in (**g**). **i** Representative fluorescent and afterglow images of mouse at different time points, post-i.v. injection of RAN1 (200 μg/mL). Scale bar: 2 cm. **j** SBR for fluorescence or afterglow in (**i**). **k** Quantification of fluorescent or afterglow intensity ratios in (**b**). ***$P < 0.001$. Data are presented as mean values ± s.d. (**$P < 0.01$, $n = 3$). Statistical differences were analyzed by Student's $t$ test. Source data are provided as a Source Data file.

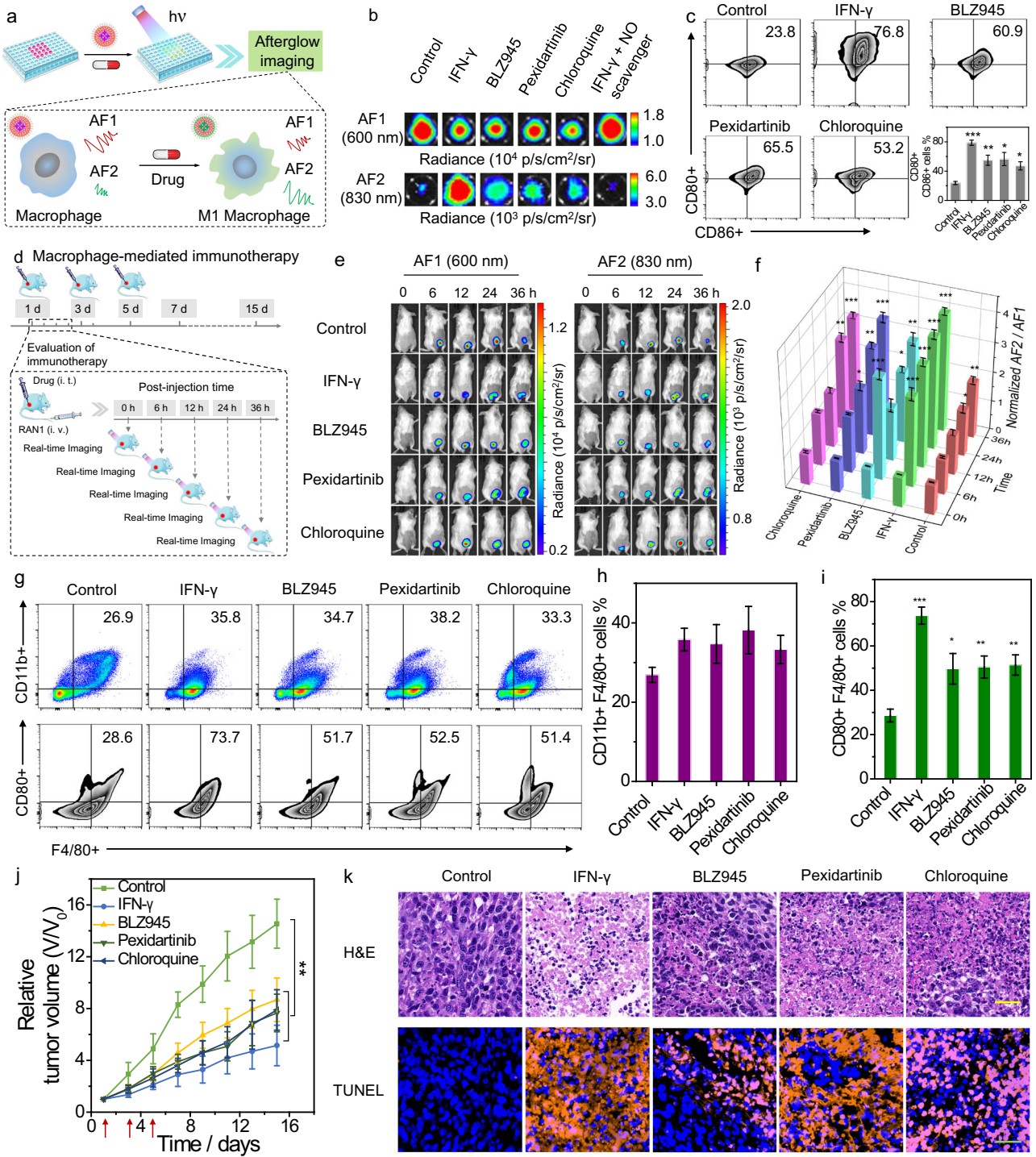

**Fig. 5 Afterglow imaging of macrophage-modulated immunotherapy. a** Schematic diagram of RAN1 for imaging modulator-stimulated macrophage polarization. **b** Representative afterglow images of RAW264.7 macrophages treated with different modulators and then with RAN1 (10 μg/mL). **c**, Flow cytometric analysis of RAW264.7 macrophages treated with different modulators and the corresponding percentages of CD86+ and CD80+. Data are presented as mean values ± s.d. (***$P < 0.001$, **$P < 0.01$, *$P < 0.05$, $n = 3$). Statistical differences were analyzed by Student's $t$ test. **d** Schematic illustration of the procedure of modulator administration and fluorescence or afterglow imaging. **e** Representative afterglow images of 4T1 tumor-bearing mice treated with modulators (i.t.) and then i.v.-injected with RAN1 (200 μg/mL). Scale bar: 2 cm. **f** The corresponding quantification of afterglow intensity ratios (AF2/AF1) in (**e**). **g** Flow cytometric analysis showed the expression of TAM marker (CD11b) and M1-phenotype macrophage marker (CD80) in tumors. The corresponding percentages of **h**, CD11b+ F4/80+ cells and **i**, CD80+ F4/80+ cells in tumors. **j** Tumor growth profiles of mice treated with different modulators. Data are presented as mean values ± s.d. (***$P < 0.001$, **$P < 0.01$, *$P < 0.05$, $n = 3$). Statistical differences were analyzed by Student's $t$ test. **k** Representative H&E- and TUNEL-stained tumor slices of each group on the second day post injection. The sections stained with orange represented the apoptotic cells. Scale bar: 100 μm. Each experiment was repeated for three times. Source data are provided as a Source Data file.

Fig. 31). Specifically, IFN-γ, a classic macrophage polarization modulator, can stimulate proinflammatory macrophages and induce NO diffuses[44]. BLZ945 and pexidartinib are inhibitors of colony-stimulating factor 1 receptor (CSF1R), and the inhibition of CSF1R can eliminate or repolarize macrophages in the TME[43]. Chloroquine, a proven antimalarial drug, can serve as an anti-tumor immune modulator that switches TAMs from M2 to tumor-killing M1 phenotype[45]. Those modulators could induce macrophage polarization to switch TAMs toward tumor-killing M1 phenotype and thereby affect the endogenous NO level. As shown in Fig. 5b and Supplementary Fig. 31, RAW264.7 macrophages incubated with IFN-γ, BLZ945, pexidartinib, or chloroquine showed enhanced ratios for both AF2/AF1 and FL2/FL1, compared with that treated with PBS. Moreover, the incubation of an additional nitric oxide scavenger (Carboxy-PTIO) could significantly reduce the ratios for both fluorescence and afterglow in contrast to IFN-γ alone. Furthermore, the normalized after-glow intensity ratio (AF2/AF1) showed a higher degree of response than the fluorescence intensity ratio (FL2/FL1) (Supplementary Fig. 32), which could be attributed to the lower background signals of afterglow from cellular culture condition. These results indicated that RAN1 could sensitively and specifically image the variation of NO during the macrophage polarization, using ratiometric afterglow imaging.

After treatment with these modulators, the expression of CD86 and CD80 (M1-phenotype macrophage markers) was analyzed by flow cytometry[46,47] (Fig. 5c). Notably, those modulators could obviously upregulate the expression of CD80 and CD86, compared with the PBS-treated group, and IFN-γ induced the highest degree of macrophage polarization among all four modulators. In addition, the immunofluorescence staining of iNOS (one of the markers for M1-phenotype macrophage and a key enzyme for NO generation) in RAW264.7 cells upon incubation with different polarization modulators have shown that the expression level of iNOS in RAW264.7 cells incubated with IFN-γ was significantly higher than other polarization modulators (Supplementary Fig. 33), which is consistent with the NO level quantified by ratiometric afterglow imaging and level of M1-phenotype macrophage markers (CD86 and CD80) determined by flow data. These results further confirmed that the real-time imaging of NO released by M1 macrophages can serve as a valid parameter for the evaluation of polarization degree.

We next applied RAN1 for real-time afterglow imaging of NO in macrophage-modulated immunotherapy by administering IFN-γ, BLZ945, pexidartinib, and chloroquine to activate macrophage polarization (Fig. 5d). After intratumoral injection of these modulators, 4T1 tumor-bearing mice were i.v.-injected with RAN1, and then the fluorescent and afterglow images were longitudinally recorded and quantified (Fig. 5e and Supplementary Fig. 34a). The higher intensity ratios for both afterglow and fluorescence were observed for modulators-treated mice compared to control mice, indicating that those modulators could induce a higher level of endogenous NO. Specifically, at 36 h post injection in the first modulators administration, the fluorescence intensity ratios (FL2/FL1) were 1.41 (control), 1.64 (IFN-γ), 1.54 (BLZ945), 1.57 (pexidartinib), and 1.56 (chloroquine) for fluorescence (Supplementary Fig. 34b), respectively. In contrast, the afterglow intensity ratios (AF2/AF1) were 1.86 (control), 3.98 (IFN-γ), 2.87 (BLZ945), 3.39 (pexidartinib), and 3.27 (chloroquine) for afterglow (Fig. 5f), which validated that afterglow had the higher sensitivity compared to in vivo fluorescence imaging.

After separation of the tumor tissues, ex vivo flow cytometric analysis was performed to provide further mechanistic validation for results obtained using afterglow imaging in vivo (Supplementary Fig. 35). The gating strategy for quantification of these biomarkers is shown in Supplementary Fig. 36. As shown in

Fig. 5g, all tumors treated with macrophage polarization modulators showed the presence of tumor-associated macro-phages with increased M1-phenotype markers (CD80+ F4/80+). Especially, the tumor treated with IFN-γ showed the highest percentage of M1 markers among all groups (Fig. 5h, i), corresponding to the highest afterglow intensity ratio AF2/AF1.

Next, we recorded the tumor volumes and body weights after various treatments every other day. The tumor growth curves showed that mice treated with modulators displayed significantly lower tumor volumes than the control group, validating the stronger inhibition ability of tumor growth (Fig. 5j). The histological hematoxylin and eosin (H&E) staining results clearly revealed that the tumors treated with those four modulators had many more necrotic areas than the control group (Fig. 5k). Furthermore, ex vivo sections stained with terminal deoxynu-cleotidyl transferase-dUTP nick end labeling (TUNEL) revealed more apoptotic tumor cells in the modulator-treated groups, compared with the control group (Fig. 5k and Supplementary Fig. 37). These results validated the good correlation among ratiometric afterglow intensity, macrophage polarization and anticancer effects in vivo. Body weight of mice was recorded during cancer treatment and showed no significant fluctuations for modulator-treated mice in contrast to the control group (Supplementary Fig. 38). After imaging or therapy, major organs of mice from each group were collected for H&E staining, and negligible pathological changes were observed (Supplementary Figs. 39 and 40), suggesting the excellent biocompatibility of RAN1 in vivo. The above results demonstrated the potential of RAN1 as a promising visualization tool for predicating the macrophage polarization and screening modulators for macrophage-modulated immunotherapy.

## Discussion

Currently, molecular imaging strategies such as bioluminescence[48], chemiluminescence[49], and afterglow luminescence can eliminate the requirement for spontaneous light irradiation, which have attracted the increasing interest. Most of bioluminescent probes are enzyme-dependent and chemiluminescent probes are based on flash-type Schaap's dioxetanes or Luminols[50,51]. And those biolu-minescent or chemiluminescent probes showed the uncontrollable release of photons, which make it difficult to accurately detect biotargets in highly heterogeneous or dynamic biological scenarios. Interestingly, afterglow luminescence is externally controllable release of photons due to the separation of laser irradiation and photons acquisition, which holds great promise for sensitive and noninvasive imaging of biomolecules in living subjects[13,22]. How-ever, up to now, the application of afterglow probes for molecular imaging has suffered from unreliable quantification of biotargets, mainly owing to the following limitations: (1) inevitable attenuation of afterglow intensity over time after light cessation, resulting in the failed acquisition of generated afterglow signal; (2) structural inertness of current afterglow materials, making it difficult to cus-tomize activatable afterglow probes to indicate the level of mole-cular targets via outputting a "turn-on" or "ratiometric" afterglow signal; (3) lack of appropriate probe design strategy, compromising the development of a universal afterglow sensing platform; and (4) the interference of factors, such as laser parameters, penetration depth, and probe concentration for afterglow, reducing the relia-bility of afterglow signal.

Inspired by FRET, we addressed these issues with the design of a universal ratiometric afterglow nanoplatform (RAN) based on a de novo ARET strategy for sensing and imaging of specific biotargets. Thus, the ARET-based ratiometric probes not only overcome the attenuation of afterglow intensity, eliminating the interference of other factors, but also exhibit a higher imaging

reliability in vivo and SBR, both of which make RAN more promising for reliable quantitative analysis. Theoretically, we can customize various ratiometric afterglow imaging probes by introducing responsive units that matches the spectrum of the afterglow substrate. For instance, by taking the afterglow substrate and afterglow initiator as an afterglow generation unit, coupled with introducing a responsive molecular probe as the detection unit, we can achieve the quantitative detection of analytes like NO, ONOO$^-$, and pH, based on the afterglow emission ratio (AF2/AF1), as demonstrated in this work, thus confirming the universality and feasibility of RAN for the design of activatable afterglow probes. Because of the significantly improved SBR caused by excitation-free afterglow imaging, RAN can accurately reflect the level of analyte changes, which is critical in the engineering of activatable specific detection and reliable imaging.

Targeting tumor-associated macrophages (TAMs) is considered a promising strategy for tumor immunotherapy[46,52], which is dependent on the immune modulation of TAMs to an anti-tumorigenic M1 phenotype to achieve anticancer efficacy[53]. Thus, the ability to measure immune response in real-time is critical to fully comprehend the interaction between macrophage and cancer cells and distinguish responders from non-responders, which is essential for the development of macrophage-modulated immunotherapy[41,54,55]. Therefore, we then applied RAN1 for evaluation of macrophage-modulated immunotherapy by real-time imaging of M1-phenotype macrophage-released proinflammatory cytokines, such as NO, a hallmark for the polarization of TAMs to the M1 phenotype[42]. Using RAN1, we enabled a noninvasive and real-time evaluation of NO level in solution and in mice with a substantial improvement of reliability, owing to its built-in correction capability and high SBR. This means that the ratiometric afterglow nanoplatform easily facilitates noninvasive assessment of macrophage-modulated immunotherapy, or high-throughput screening of immunotherapeutic modulators in living animals, for future applications.

## Methods

**Ethical statement**. All animal procedures were performed in accordance with the Guidelines for the Care and Use of Laboratory Animals of Hunan University, and experiments were approved by the Animal Ethics Committee of the College of Biology (Hunan University).

**Materials and characterization**. All chemicals were purchased from commercial suppliers and used without further purification. Poly[2-methoxy-5-(2-ethylhexyloxy)-1,4-phenylenevinylene] (MEHPPV) was obtained from Xi'an Polymer Light Technology Corp. Acetaminophen, BLZ945, pexidartinib and chloroquine were obtained from Shanghai Bide Pharmaceutical Technology Co., Ltd. LysoTracker Green and IFN-γ were obtained from Beyotime Life Technologies Co., Ltd. DMEM, FBS, and penicillin–streptomycin were purchased from Gibco, Life Technologies. CD80, CD86, CD11b, and F4/80 antibodies were purchased from Invitrogen. The One Step TUNEL Apoptosis Assay Kit (Red) was obtained from Wuhan Saville Biotechnology Co., Ltd. Reactive oxygen/nitrogen species were prepared according to the previously reported literature. Mice were purchased from Hunan Slake Jingda Laboratory Animal Co., Ltd.

Thin-layer chromatography (TLC) was conducted using silica gel 60 F254, and column chromatography was carried out over silica gel (200–300 mesh) obtained from Qingdao Ocean Chemicals (Qingdao, China). Mass spectra were performed using an LCQ Advantage ion trap mass spectrometer (Thermo Finnigan). NMR spectra were recorded on a Bruker DRX-400 spectrometer using TMS as an internal standard. Transmission electron microscope images were accomplished using a JEM-2100 instrument (JEOL). Dynamic light scattering (DLS) measurements were made on a Malvern Zetasizer Nano ZS90 (Malvern). Water was purified and doubly distilled by a Milli-Q system (Millipore, USA). UV-Visible absorption spectra were acquired via the Shimadzu UV-2600 UV-VIS-NIR spectrophotometer. Fluorescence spectra were recorded on a HITACHI F4600 fluorescence spectrophotometer with a 1-cm standard quartz cell. The fluorescent and afterglow images of centrifuge tube or mice were obtained via an IVIS Lumina XR Imaging System (Caliper, USA) equipped with a cooled charge-coupled device (CCD) camera. Fluorescent images of cells were obtained from the Olympus FV1000-MPE laser scanning confocal microscope (Japan).

**Synthesis of BDP**. BODIPY (480 mg, 1 mmol) and benzaldehyde (265 mg, 2.5 mmol) were dissolved in 5 mL anhydrous acetonitrile under $N_2$ atmosphere in a two-neck flask. Then piperidine (0.5 mL) and glacial acetic acid (0.3 mL) were introduced into the flask through a syringe. The reaction mixture was stirred at 85 °C for 4 h. Then the solvent was removed under reduced pressure and purified by flash column chromatography over silica gel using dichloromethane/hexane as the eluent to obtain the desired product BDP as a dark-blue solid. The yield is 37%. $^1$H NMR (400 MHz, CDCl$_3$) δ 8.14 (d, J = 16.7, 2H), 7.75 (d, J = 16.7, 2H), 7.68 (d, J = 7.3, 4H), 7.57–7.52 (m, 3H), 7.43 (t, J = 7.4, 4H), 7.36 (dd, J = 8.3, 6.3, 2H), 7.33–7.28 (m, 2H), 1.42 (s, 6H). $^{13}$C NMR (101 MHz, CDCl$_3$) δ 143.76, 136.69, 135.13, 134.63, 132.07, 129.99, 127.43, 124.86, 124.72, 124.58, 124.08, 123.48, 123.01, 113.39, 9.01. MALDI-MS (ESI): calculated for C$_{33}$H$_{25}$BBr$_2$F$_2$N$_2$ 657.04, [M + H]$^+$, found 657.93.

**Synthesis of NRM**. To a 10-mL sealed tube was added NRM-NO$_2$ (253 mg, 0.2 mmol), iron powder (112 mg, 2 mmol), and AcOH (10 mL). The reaction mixture was heated to 100 °C for 6 h and then cooled to room temperature. The reaction was neutralized with saturated sodium bicarbonate solution and extracted with EtOAc. The combined organic layers were washed with water (50 mL), dried over anhydrous Na$_2$SO$_4$ and evaporated in vacuo. The residue was purified by recrystallization from petroleum ether and ethyl acetate to obtain NRM as a russet solid. The yield is 50%. $^1$H NMR (400 MHz, $d_6$-DMSO) δ 7.70 (q, J = 8.1 Hz, 12H), 7.60 (d, J = 13.3 Hz, 8H), 7.16 (d, J = 11.4 Hz, 4H), 7.06 (d, J = 7.8 Hz, 8H), 6.52 (d, J = 15.9 Hz, 4H), 5.99 (s, 4H), 4.18 (q, J = 6.8 Hz, 8H), 1.26 (t, J = 7.0 Hz, 12H). $^{13}$C NMR (101 MHz, DMSO) δ = 166.85, 150.40, 148.60, 145.52, 144.17, 143.31, 140.59, 135.89, 130.38, 129.28, 127.24, 126.30, 123.68, 116.90, 102.07, 60.38, 14.70. MALDI-MS (ESI): calculated for C$_{70}$H$_{60}$N$_6$O$_8$S$_3$ 1208.36 [M]$^+$, found 1208.22.

**Synthesis of IR780**. 2-Chloro-1-formyl-3-(hydroxymethylene) cyclohex-1-ene (2.0 g, 11.4 mmol), anhydrous sodium acetate (2.0 g) and 1,2,3,3-tetramethyl-3H-indolium iodide (6.87 g, 23 mmol) were added into acetic anhydride (20 mL) solution. The mixture was was heated at 60 °C for 1 h. Then the mixture was cooled to room temperature and filtered. The solid was washed with saturated NaHCO$_3$ buffer until no bubble appeared. Then the solid was washed with water twice. After that, the solid dried under vacuum to afford light-green solid. The yield is 55%. $^1$H NMR (400 MHz, $d_6$-DMSO) δ 8.25 (d, J = 14.2 Hz, 2H), 7.64 (t, J = 7.0 Hz, 2H), 7.47–7.41 (m, 4H), 7.29 (ddd, J = 8.1, 5.6, 2.8 Hz, 2H), 6.31 (d, J = 14.2 Hz, 2H), 3.70 (s, 6H), 2.72 (t, J = 5.5 Hz, 4H), 1.91-1.81 (m, 2H), 1.67 (s, 12H). $^{13}$C NMR (101 MHz, $d_6$-DMSO) δ 173.09, 148.14, 143.31, 143.14, 141.45, 129.01, 126.53, 125.59, 122.85, 111.89, 102.35, 49.34, 32.04, 27.79, 26.36.

**Synthesis of ORM**. 3-Mercaptopropionic acid (130 μL, 1.5 mmol) and triethylamine (210 μL, 1.5 mmol) were added to a solution of IR780 iodide (0.75 g, 1.3 mmol) in DMF (5 mL). The mixture was stirred at room temperature for 20 h, dichloromethane was added, and the whole organic phase was washed with brine. The organic extracts were dried over Na$_2$SO$_4$, filtered, and evaporated. The crude product was purified by recrystallization from 2-propanol to afford ORM as dark red crystals. The yield is 59%. $^1$H NMR (400 MHz, $d_6$-DMSO) δ 8.71 (d, J = 14.1 Hz, 2H), 7.59 (d, J = 7.4 Hz, 2H), 7.41 (d, J = 3.5 Hz, 4H), 7.31–7.22 (m, 2H), 6.28 (d, J = 14.2 Hz, 2H), 3.66 (s, 6H), 2.94 (t, J = 7.1 Hz, 2H), 2.65 (s, 2H), 2.40 (t, J = 7.0 Hz, 2H), 1.81 (s, 1H), 1.68 (s, 13H). $^{13}$C NMR (101 MHz, $d_6$-DMSO) δ 172.74, 145.11, 143.41, 141.29, 133.29, 128.92, 125.22, 122.77, 111.56, 101.98, 49.12, 45.89, 31.78, 27.71, 26.16. MALDI-MS (ESI): calculated for C$_{35}$H$_{41}$N$_2$O$_2$S$^+$ 553.28 [M]$^+$, found 553.13.

**Synthesis of CS**. Fisher aldehyde (201 mg, 1 mmol) and 9-(2-carboxyphenyl)-6-(diethylamino)-1,2,3,4-tetrahydroxanthylium (376 mg, 1 mmol) were dissolved in acetic anhydride (10 mL), and the reaction mixture was heated to 50 °C and further stirred for 30 min. Then, water (15 mL) was added to the reaction mixture to quench the reaction. The solvent was removed under reduced pressure to give the crude product, which was purified by silica gel flash chromatography using CH$_2$Cl$_2$/ethanol (200:1 to 20:1) as eluent to afford compounds CS as black solid. The yield is 35%. $^1$H NMR (400 MHz, $d_6$-DMSO) δ 12.79 (s, 1H), 8.56 (d, J = 14.2 Hz, 1H), 8.15 (dd, J = 7.8, 0.8 Hz, 1H), 7.80 (td, J = 7.5, 1.2 Hz, 1H), 7.69 (dd, J = 12.6, 4.5 Hz, 2H), 7.50-7.43 (m, 2H), 7.31 (dd, J = 9.3, 4.5 Hz, 2H), 6.81 (dd, J = 9.2, 2.4 Hz, 1H), 6.70 (d, J = 2.3 Hz, 1H), 6.62 (d, J = 9.1 Hz, 1H), 6.27 (d, J = 14.1 Hz, 1H), 3.73 (s, 3H), 3.53 (q, J = 7.0 Hz, 4H), 2.66 (t, J = 5.6 Hz, 2H), 2.34-2.16 (m, 2H), 1.74 (t, J = 14.2 Hz, 8H), 1.18 (t, J = 7.0 Hz, 6H). $^{13}$C NMR (101 MHz, $d_6$-DMSO) δ 174.43, 172.47, 167.17, 162.19, 155.47, 151.75, 151.23, 143.25, 141.45, 135.66, 133.31, 131.21, 130.75, 129.95, 129.83, 129.00, 128.16, 125.72, 122.89, 120.60, 114.84, 113.07, 112.54, 111.96, 96.06, 55.37, 49.55, 44.82, 32.08, 28.18, 26.78, 21.52, 20.56, 12.80.

**Synthesis of PRM**. Hydrazine hydrate (3 mmol) and BOP (0.5 mmol) were added to a solution of CS (0.5 mmol) in dry 1,2-dichloroethane. The mixture was vigorously stirred at room temperature for 2 h, and the solvent was evaporated under reduced pressure. The residue was purified by a silica gel column using CH$_2$Cl$_2$: ethanol (v/v, 20:0 to 20: 1) to afford compound PRM as a yellow solid. The yield is

62%. $^1$H NMR (400 MHz, $d_6$-DMSO) δ 7.73 (d, J = 7.3 Hz, 1H), 7.53 (t, J = 7.1 Hz, 1H), 7.47 (t, J = 7.2 Hz, 1H), 7.38 (d, J = 12.6 Hz, 1H), 7.27 (d, J = 7.2 Hz, 1H), 7.13 (dd, J = 10.0, 7.7 Hz, 2H), 6.79 (dd, J = 12.6, 7.6 Hz, 2H), 6.34 (dd, J = 8.8, 2.3 Hz, 1H), 6.24 (dd, J = 11.6, 5.6 Hz, 2H), 5.41 (d, J = 12.7 Hz, 1H), 4.38 (s, 2H), 3.33 (d, J = 7.4 Hz, 4H), 3.14 (s, 3H), 2.50 (d, J = 1.6 Hz, 2H), 1.78-1.71 (m, 1H), 1.64 (d, J = 5.1 Hz, 6H), 1.51–1.44 (m, 1H), 1.36–1.20 (m, 2H), 1.10 (t, J = 6.9 Hz, 6H). $^{13}$C NMR (101 MHz, $d_6$-DMSO) δ 165.74, 157.41, 152.82, 150.06, 148.46, 147.39, 145.47, 138.63, 132.70, 131.10, 128.72, 128.16, 127.99, 123.63, 122.50, 122.06, 120.19, 119.61, 119.18, 108.61, 106.52, 105.47, 104.47, 97.45, 92.49, 66.96, 45.31, 44.16, 29.35, 28.45, 25.18, 22.88, 22.14, 12.85. MALDI-MS (ESI): calculated for $C_{37}H_{40}N_4O_2$ 573.31 $[M + H]^+$, found 573.24.

**Preparation of RAN1**. All the nanoparticles were prepared using the amphiphilic polymer-assisted nanoprecipitation method[5,25]. For RAN1, a tetrahydrofuran (THF) solution (2 mL) containing NRM (0, 0.025, 0.05, 0.1, and 0.2 mg), MEHPPV (0.25 mg), TPP (0.002 mg) or BDP (0.002 mg), and Pluronic F127 (20 mg) was rapidly injected into distilled-deionized water (10 mL) under sonication. After sonication for another 10 min, the solution was evaporated at 50 °C by rotary evaporation to remove excess THF. Finally, the RAN1 solution was purified by ultrafiltration (10 K, 4000 rpm) several times. The final concentration of RAN1 was determined by the concentration of NRM.

**Preparation of RAN2 and RAN3**. For preparation of RAN2, a mixed tetrahydrofuran (THF) solution (2 mL) containing ORM (25 μg), MEHPPV (0.25 mg), TPP (0.05 mg), PSMA (3 mg), and Pluronic F127 (20 mg) was rapidly injected into distilled-deionized water (10 mL) under sonication. For RAN3, a mixed tetrahydrofuran (THF) solution (2 mL) containing PRM (0.2 mg), MEHPPV (0.25 mg), BDP (1 μg), and PSMA-PEG (10 mg) was rapidly injected into distilled-deionized water (10 mL) under sonication. After sonication for another 10 min, the solution was evaporated at 50 °C by rotary evaporation to remove excess THF. Finally, the RAN1 solution was purified by ultrafiltration (10 K, 4000 rpm) several times. The final concentration of RAN2 was determined by the concentration of ORM.

**Fluorescence and afterglow imaging in solution**. Experiments to measure fluorescence and afterglow were performed in PBS (10 mM) buffer solutions. For fluorescence imaging, the fluorescent images were acquired on an IVIS Spectrum imaging system under fluorescence mode with an acquisition time of 0.1 s and excitation wavelength at 500 nm. For afterglow imaging, those samples were pre-illuminated with a 660-laser (0.80 W/cm$^2$) or white light (0.4 W/cm$^2$) for 30 s. Immediately after irradiation, the afterglow images were obtained on an IVIS Spectrum imaging system under bioluminescence mode with an acquisition time of 30 s, equipped with DsRed emission filter (550-650 nm) for MEHPPV and ICG emission filter (800–875 nm) for NRM, ORM or PRM. In attenuation studies, RAN1 (20 μg/mL NRM, 50 μg/mL MEHPPV) was pre-irradiated with a 660-nm laser (0.80 W/cm$^2$) for 30 s, and then the afterglow signals were collected every 10 s. The afterglow intensity in each image was quantified by applying a region of interest (ROI) over the image, using the Lumina XR Living Image software, version 4.3.

**Imaging of NO in the macrophage polarization process in vitro**. 4T1 or RAW264.7 cells were maintained in RPMI-1640 medium with 10% fetal bovine serum (FBS, GIBCO) and 1% penicillin–streptomycin at 37 °C in a humidified atmosphere containing 20% O$_2$ and 5% CO$_2$ as the normoxic condition. 5000 RAW264.7 cells per well were seeded in a 96-well plate and incubated for 24 h in a humidified incubator for adherence. Then the RAW264.7 macrophages were polarized in M1-phenotype macrophages by incubation with different macrophage polarization modulators, including 20 ng/mL of IFN-γ, 5 μg/mL BLZ945, 5 μg/mL pexidartinib, and 12.5 μg/mL chloroquine, in RPMI-1640 medium. Then RAN1 (10 μg/mL NRM, 25 μg/mL MEHPPV) was incubated with those cells for 24 h. The fluorescent images of these cells were acquired on an IVIS Spectrum imaging system under fluorescence mode with an acquisition time of 0.1 s. Immediately after irradiation for 30 s (660 nm laser, 0.80 W/cm$^2$), the afterglow images were obtained on an IVIS Spectrum imaging system under bioluminescence mode with an acquisition time of 30 s.

**Tissue-penetration studies**. RAN1 (20 μg/mL) was pre-irradiated with a 660 nm laser (0.80 W/cm$^2$) for 30 s in the presence of NO (50 μM), or not. Solutions were then placed under chicken tissues of varying thickness (0, 0.1, 0.2, 0.4, and 0.6 cm). The afterglow images were acquired for 30 s with DsRed filter for MEHPPV and ICG filter for NRM-NO after removal of the laser. The fluorescent images were acquired for 1 s at 600 nm and 830 nm with excitation at 500 nm.

**Cytotoxicity assay**. 4T1 cells and RAW264.7 cells were purchased from Cell Bank of the Chinese Academy of Sciences. Cytotoxicity assays were carried out using 4T1 cells. Cell viability was determined using the CCK-8 assay. IN total, 5000 cells per well were seeded in a 96-well plate and incubated for 12 h in a humidified incubator for adherence. RAN solution was then added to cells at the final concentration of 0, 5, 10, 20, 30, 40, and 50 μg/mL and then incubated for 24 h. CCK-8

reagent diluted by RPMI-1640 (FBS free) medium (10%) was added to each well after the removal of culture media and incubated for 0.5 h. Following that, the absorbance was measured at 450 nm on a plate reader Synergy 2 Multi-Mode Microplate Reader (Bio-Tek, Winooski, VT).

**Flow cytometric analysis of cells**. RAW264.7 macrophages were seeded at a density of $8 \times 10^4$ cells/well. Next, the cells were treated with polarization modulators, including 25 ng/mL IFN-γ, 5 μg/mL BLZ945, 5 μg/mL pexidartinib, and 12.5 μg/mL chloroquine, for 24 h. Anti-mouse CD16/32 (Fc block, clone 2.4 G, BD Biosciences) was pre-added to block nonspecific binding of immunoglobulin to macrophage Fc receptors. Then the cells were then washed with PBS, centrifuged and stained for anti-CD86-PE (0.125 μg) and anti-CD80-APC (0.06 μg) at 4 °C for 30 min. The concentration at each antibody was used as recommended by the manufacturer. Post-staining, those cells were quantified by flow cytometry analysis (BD C6 Plus).

**Immunofluorescence Assays**. RAW264.7 cells were seeded on 20-mm confocal dish. Next, the cells were incubated with polarization modulators, including 25 ng/mL IFN-γ, 5 μg/mL BLZ945, 5 μg/mL pexidartinib and 12.5 μg/mL chloroquine for 24 h. Then, those cells were fixed in 4% paraformaldehyde for 30 min at room temperature, followed by permeabilization with 0.2% Triton X-100 in DPBS for 5 minutes. Those cells were blocked for nonspecific binding with 5% BSA in DPBS, followed by incubating with Anti-iNOS antibody (Abcam, ab283655, 1:100) at room temperature for 5 h. After washing, those cells were stained with secondary antibody (Alexa Fluor 488-labeled Goat Anti-Rabbit IgG(H + L), Beyotime) for 1 h at room temperature and was stained with Hoechst 33258 (100 nmol, Beyotime) for 30 min. The figures were analyzed by Olympus FV1000 laser confocal microscopy (Japan).

**In vivo experiment**. All animal procedures were performed in accordance with the Guidelines for the Care and Use of Laboratory Animals of Hunan University, and experiments were approved by the Animal Ethics Committee of the College of Biology (Hunan University). Fluorescent images of mice were acquired on an IVIS Spectrum imaging system under fluorescence mode with an acquisition time of 0.1 s. For afterglow imaging, the mice were pre-illuminated with a 660 nm laser (1.0 W/cm$^2$) for 60 s. Immediately after removing the laser, the afterglow images were acquired on an IVIS Spectrum imaging system with an acquisition time of 60 s, equipped with a DsRed emission filter for MEHPPV and ICG emission filter for NRM-NO.

**Flow cytometric analysis and immunofluorescence staining of iNOS**. The $Nos2^{-/-}$ and WT mice were the first i.p. injected with 100 μL of 1 mg/mL LPS or 100 μL PBS for 6 h. Then the mice were sacrificed by dislocation, and the abdominal cavity was opened. The prepared perfusate (0.05% collagenase IV in PBS) was infused through the hepatic portal vein until the liver completely turned yellowish-white, and the livers were harvested after perfusion. Part of isolated liver tissue slices was acquired for immunofluorescence staining, using an Anti-iNOS antibody (Abcam, ab283655, 1:100). And part of the liver is minced and incubated with 0.05% collagenase IV for 1 h at 37 °C. The cell suspension was passed through a 70-μm filter to remove debris and to obtain a single-cell suspension. The obtained liver cell suspension was centrifuged (300 × g, 5 min), the obtained precipitate was washed with PBS, and then centrifuged (50 × g, 3 min). Then, the supernatant was collected and centrifuged (300 × g, 5 min) to obtain nonparenchymal cells, which were seeded at a density of $1 \times 10^6$ cells/group, fixed in 4% paraformaldehyde, permeabilized with 0.2% Triton X-100, blocked with 5% BSA, followed by incubating with 0.5 mL Anti-iNOS antibody (Abcam, ab283655, 1:100) at room temperature for 5 h. After washing, those cells were stained with a secondary antibody (Alexa Fluor 488-labeled Goat Anti-Rabbit IgG(H + L), Beyotime) for 1 h at room temperature. Finally, the samples were washed twice to remove unbound antibodies and analyzed using flow cytometry.

**Imaging of acetaminophen (APAP)-induced liver injury in BALB/c**. 10 female BALB/c (18–20 g) mice were divided into two groups on average: PBS group: intraperitoneal-injected PBS and intravenous injection RAN2 and APAP group: intraperitoneal-injected APAP and intravenous injection RAN2. Mice were intraperitoneally injected with 100 μL of 200 mg/kg acetaminophen (APAP) or 100 μL PBS for 1 h and then intravenously injected with RAN2 (100 μL, 40 μg/mL) for 60 min. The mice were immediately imaged via an IVIS Lumina XR Imaging System. For afterglow imaging, the mice were pre-illuminated with a 660-laser (0.80 W/cm$^2$) for 60 s. Then the laser was removed, and the afterglow images were acquired on an IVIS Spectrum imaging system with an acquisition time of 60 s, equipped with a DsRed emission filter for MEHPPV and ICG emission filter for ORM. Circular ROIs were drawn over each well, and fluorescence intensity was quantified by Living Image software.

**Imaging of NO in an inflammation model**. To establish the inflammation model, mice were intradermal (i.d.) injected with 50 μL (5 mg/mL) of lipopolysaccharide into the dorsal skin of the rear paw. For imaging of LPS-induced inflammation, mice bearing LPS-induced inflammation were i.d.-injected with 25 μL RAN1 (10 μg/mL NRM, and 25 μg/mL MEHPPV) into the dorsal skin of the rear paw.

For the sc tumor model, 4T1 cells ($10^6$) suspended in 25 μL of PBS were sc-injected into the back of each BALB/C mouse (7–8 weeks, female). For imaging endogenous NO within the tumor, mice bearing 4T1 xenograft tumor were then i.v.-injected with RAN1 200 μL RAN1 (200 μg/mL NRM, 500 μg/mL MEHPPV). After administration, mice were anaesthetized using rodent ventilator with 2 L/min air with 4% isoflurane. The fluorescent images of mice were acquired on an IVIS Spectrum imaging system under fluorescence mode. Immediately after irradiation for 60 s (660 nm laser, 0.80 W/cm$^2$), the afterglow images were obtained on an IVIS Spectrum imaging system under bioluminescence mode with an acquisition time of 60 s.

**Comparison of the imaging of NO generation in Nos2$^{-/-}$ mice and wild-type (WT) mice.** The Nos2$^{-/-}$ mice and wild-type (WT) mice were purchased from Cyagen Biosciences (Suzhou) Inc. Those fasting Nos2$^{-/-}$ or WT mice were i.p. injected with 100 μL of 1 mg/mL LPS or 100 μL PBS, respectively, and then i.v. injected with 200 μL RAN1 (200 μg/mL NRM, 500 μg/mL MEHPPV) at 6 h post the first injection. After 60 min, the fluorescent images of mice were acquired on an IVIS Spectrum imaging system under fluorescence mode. Immediately after irradiation for 60 s (660 nm laser, 0.80 W/cm$^2$), the afterglow images were obtained on an IVIS Spectrum imaging system under bioluminescence mode with an acquisition time of 30 s. Circular ROIs were drawn over each well, and fluorescence intensity was quantified by Living Image software.

**In vivo tumor imaging, therapy, and macrophage polarization.** Mice bearing 4T1 xenograft tumor were i.t.-injected with 25 μL of PBS or different macrophage polarization modulators, including 20 μg/mL of IFN-γ, 4 mg/mL BLZ945, 4 mg/mL pexidartinib and 10 mg/mL chloroquine, and then i.v.-injected with 200 μL RAN1 (200 μg/mL NRM, 500 μg/mL MEHPPV). Then, mice were imaged by fluorescence and afterglow imaging mode. For cancer therapy, i.t. injections of various macrophage polarization modulators were performed at the first, third, and fifth day, respectively. Tumor volumes and mouse body weights were measured every other day during the 15-day study. Tumors were collected on the second day, and major organs were collected on the 15th day post-treatment for hematoxylin and eosin (H&E) staining. The tumor volumes were calculated according to the following formula: width$^2$ × length/2.

For TUNEL staining, representative tumors from each group were harvested at 24 h post injection of modulators. Tumors were frozen in OCT and sliced into 5-μm thin sections, stained with One Step TUNEL Apoptosis Assay Kit (Red, Wuhan Servicebio Technology Co., Ltd.), according to the manufacturer's protocol, imaged using a Digital slice scanning system (Pannoramic MIDI), and analyzed with CaseViewer software.

For isolation of tumor cells and flow cytometric analysis, 24 h post-i.t. injection of different modulators, representative tumors from each group were harvested, minced, and treated with 1 mg/ml type I collagenase and 0.1 mg/ml of DNase I, followed by incubation for 1 h at 37 °C and 5% CO$_2$. The cell suspension was passed through a 70 μm filter to remove debris and to obtain a single-cell suspension. Then, the tumor cells were preincubated (30 min, 4 °C) with anti-CD16/32 monoclonal antibody (0.5 μg, Fc block, clone 2.4 G, BD Biosciences) to block nonspecific binding and then stained (30 min, 4 °C) with appropriate dilutions of various combinations of the following fluorochrome-conjugated antibodies: anti-CD11b-Alexa Fluor 488 (0.25 μg, clone M1/70), anti-F4/80-PE-Cy5 (0.25 μg, clone BM8), and anti-CD80-APC (0.06 μg, clone 16-10A1). Post-staining, the samples were washed twice to remove unbound antibodies and analyzed using flow cytometry.

**Statistics and reproducibility.** All data were expressed in this article as mean result standard deviation (SD). All data were analyzed using Student's t tests. Data analysis was performed using GraphPad software. Reported P values were two-sided and considered significant when lower than 0.05.

**Reporting summary.** Further information on research design is available in the Nature Research Reporting Summary linked to this article.

## Data availability
The authors declare that all other data related to this study are available in the article/and or its Supplementary Information files. A reporting summary for this article is available as an Additional Information file. Source data are provided with this paper.

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

## Acknowledgements

This work was supported by the National Key R&D Program of China (2019YFA0210100) and the National Natural Science Foundation of China (grants, 21804039, 51872088, 21977027, 21890744, and 21890744). The Science and Technology Project of Hunan Province (2021RC2051), China Postdoctoral Science Foundation (2021M690958).

## Author contributions

Y.L., L.T., G.S., X.-B.Z., and W.T. conceived and designed experiments; Y.L. and L.T. performed the experiments; Y.L., G.S., Y.L., and X.-B.Z. performed data analysis and wrote the manuscript. All authors discussed the results and commented on the manuscript.

## Competing interests

The authors declare no competing interests.
