## [Peer Review File · Nature Communications]

Reviewers' Comments:

Reviewer #1:

Remarks to the Author:

In this work, a ratio imaging nano-platform based on afterglow resonance energy transfer (ARET) is developed, which is used to customize various activatable afterglow probes for reliable quantification and molecular analysis of specific analytes Imaging. Authors successfully designed the first NO response ratio afterglow nanoprobe RAN1. The nanoprobe is able to evaluate the polarization degree of macrophages in the course of cancer immunotherapy, and provide a reliable way to predict the effect of immunotherapy. We believe the main idea of this work is very meaningful, but there are still some scientific issues to be clarified. A revision is required to improve this work for publication in this journal. Our comments and questions are list as follow.

- 1.The shortages of traditional fluorescence imaging are introduced in the introduction. However, recent works have indicated that there are some optical imaging strategies that can overcome these shortcomings, such as chemiluminescence imaging (e.g. 10.1002/anie.202007649), bioluminescence imaging (e.g. 10.1038/s41467-020-18051-1), and afterglow imaging (e.g. 10.1038.s41565-021-00922-3). We suggest that the authors may elaborate on these methods in the introduction part and further highlight the advantages of afterglow imaging in this work.
- 2.The ARET-based ratiometric nanoplatform proposed by authors is composed of responsive molecules (NRM, ORM or PRM), afterglow substrate (MEHPPV), surfactants (F127) and afterglow initiators (TPP or BDP) via a self-assembly strategy. What we are concerned about is the responsiveness difference between of ARET-based ratiometric nanoplatform and responsive molecules (NRM, ORM or PRM).
- 3.The imaging results illustrated in Figure 3k are different from the data in Figure 3n. As we all know, SBR compares the level of a desired signal to the level of background noise. In Figure 3n, the SBRs of AF2 signal under 0.1 cm chicken tissue and the AF1 signal under 0.6 cm chicken tissue are higher than that of FL2 under 0 cm chicken tissue. Hence, the signal should also be easily distinguished from the background in imaging results (Figure 3k). Here, we strongly recommend that the authors to zero the lower limit of the scalebar in Figure 3k and Supplementary Figure 21a to fully demonstrate the results of fluorescence imaging and make the data more consistent.
- 4.The benchmarks for data normalization in Figures 3I and Supplementary Figure 21b are not clarified.
- 5.The results in Figure 3m are puzzling. Generally speaking, the photons with longer wavelength have deeper penetration depths in bio tissues. However, FL2/FL1 in Figure show a downward trend, which means that the fluorescence emission intensity at 830 nm is more significantly affected by the depth of bio tissue than the fluorescence emission at 600nm. Authors should give a reasonable explanation for this phenomenon. On the other hand, the reason why the afterglow intensity ratio (AF2/AF1) remains constant at any depth from 0 to 0.6 cm also needs to be emphatically described. In our cognition, the attenuation rate of light intensity in a medium is basically only related to its wavelength. In Figure 3m, the significant difference between the fluorescence intensity ratio and the afterglow intensity is beyond our understanding. The authors should make it clear through further experiments or simulations.
- 6.In Figure 4b, why the AF2 signal in mouse injected with PBS stronger than that in mouse injected with LPS?
- 7.Supplementary figure 23a does not show the AF1 and FL1 signals properly after the mouse is injected with APAP.
- 8.Authors claimed that "RAN1 can monitor the fluctuations of intratumoral NO, as a biomarker of M1 macrophages". Can this process be verified under a confocal microscope? It is recommended to verify the imaging advantages of RAN1 in other scenarios compared with ratio fluorescence probes.

Reviewer #2:

Remarks to the Author:

Liu et al report the preparation and use of nanoplatforms with afterglow resonance energy transfert properties and applied them for the ratiometric in vivo quantification of diverse analytes such as NO and ONOO-.

The topic is new and of particular interest.

The reviewer has some comments :

- in the introduction, with references 7, 8 and 9 the authors should also add Liu et al, Adv Drug Deliv Rev 138 (2019) 193–210;
- do the authors have an idea of the amount of afterglow substrate, afterglow initiator, NRM, ORM or PRM incorporated into the nanoplatfrom ?
- is this amount the same whatever the responsive molecule used ?
- what is the influence of the incorporated amount on the further ARET phenomenon ?
- Fig S2 present the hydrodynamic diameter of RAN1. Is the size of RAN dependent on the responsive molecule incorporated into the nanoplatfrom ?
- in which medium was measured the hydrodynamic diameter ?
- is the nanoplatfrom stable over the time ?
- is the ARET phenomenon sensitive to the medium ?

Reviewer #3:

Remarks to the Author:

In this manuscript submission, Liu et al. developed an "afterglow" resonance energy transfer imaging system to measure nitric oxide production in vitro and in vivo. Conceivably, this approach can be used to facilitate measurement of a wide range of different physiological outputs such as ROS, pH, hypoxia and so on. Although a promising approach, this manuscript could be improved in several ways:

1. The title is non-specific and should be revised to communicate the main message without jargon.
2. The abstract is not well written. For example, a term such as "afterglow" must be defined as this is non-standard. The authors' approach is also vague in many parts of the abstract. Terms such as "M1" should be avoided here as a general reader of "imaging" technology will have no idea about this.
3. In my view, the entire manuscript requires re-writing. It is not clear who the audience is, and the authors succumb to excessive jargon throughout. The manuscript is way too long for the content.
4. Similarly, I found the figures uninformative and requiring extensive revision. Take Figure 1a as an example: it is unclear what the authors seek to communicate in the left two boxes. It is also unnecessary to introduce "macrophage-mediated immunotherapy" in this figure (which the authors are not really doing, and this term is sufficiently nebulous to create confusion – figure 1c right box is simply a fantasy of the authors). Many of the figures are far, far too small and confusing.
5. When we talk about M1 macrophages in this case, the authors would be on safer ground if they defined what they were doing in the first place vis-à-vis stimulations. While it is true that iNOS is a "marker" of an M1-like macrophage, a better approach is to state this, and then continue with the actual stimulations. For example, define that M1-like macrophages express iNOS and then proceed by defining the actual stimulation conditions. Examples of the problems with nomenclature of macrophage polarization have been published before (e.g. Murray et al. Immunity).
6. While it seems that the RAN1 approach detects NO (and peroxynitrates), the authors can't safely conclude this with the RAW cell line. Instead, at least one in vivo experiment must be done using the Nos2^{-/-} mice compared to WT. This would include the flow data (including staining for intracellular iNOS)
7. The parts on "macrophage-mediated immunotherapy" must be revised to "manipulation" or "modulation".
8. The authors would improve the manuscript by making the first figure about the chemistry of the system, in more depth.
9. Line 298. NO is not secreted; it freely diffuses.

Reviewer #4:

Remarks to the Author:

In this manuscript, the authors report a universal ARET-based ratiometric nanoplatfrom to

customize afterglow probes toward specific analytes (e.g. NO, ONOO⁻ or pH) for reliable quantification and molecular imaging. By the ratiometric afterglow imaging strategy, the challenges caused by afterglow attenuation and structural inertness of afterglow materials have been effectively addressed. Finally, the NO-responsive ratiometric afterglow probe (RAN1) is successfully applied for evaluation of macrophage-mediated immunotherapy with high reliability. I think this is a solid and important piece of work and should be of broad interest to the audience of Nature Communications. I recommend publication after addressing the following minor points :

- (1) The response mechanism of RAN2 is different from that of RAN1 and RAN3. It is an "on-off" probe, and the characteristic emission of MEHPPV at 600 nm in RAN2 is significantly lower than that of RAN1 and RAN3, is this due to the higher energy transfer efficiency between MEHPPV and ORM in RAN2, or there are other reasons?
- (2) In RAN1 characterization, the important Zeta potential is absent, please provide its Zeta potential.
- (3) RAN1 was prepared by self-assembly, will it be disassembled in vivo? What is the stability in water or biological condition? More experiments should be provided to indicate the stability of RAN1 in biological conditions.
- (4) What is the purpose of introducing these AIs? What effect does it have on the ARET process?
- (5) The UV-visible absorption and fluorescence emission spectra of NRM, ORM and PRM before and after the response should be provided.
- (6) In Fig. 4b, the afterglow signals of the rear paw are not clearly displayed. I recommend the authors to enlarge the signal area to obtain high-quality imaging pictures.
- (7) Supplementary Fig. 21d showed that the SBR of AF1 was similar to that of AF2 in in vitro experiment, however, the SBR of AF2 was significantly lower than that of AF1 in in vivo experiment (Fig. 4h). What caused the difference between in vivo and in vitro experiments?
- (8) To show the size of the mice, I recommend the authors to add a scale bar to the appropriate position on all the imaging pictures of mice, including Fig. 4b, d, g, Fig. 5e, Supplementary Fig. 22a, 23a, 28a.
- (9) The format of the references needs to be checked carefully, and some unnecessary errors should be avoided as much as possible, such as "tumor-to-Liver" in Ref. 16, "real-Time" in Ref. 40.

Point-by-point responses to the reviewers' comments:

Reviewer #1:

1. The shortages of traditional fluorescence imaging are introduced in the introduction. However, recent works have indicated that there are some optical imaging strategies that can overcome these shortcomings, such as chemiluminescence imaging (e.g. 10.1002/anie.202007649), bioluminescence imaging (e.g. 10.1038/s41467-020-18051-1), and afterglow imaging (e.g. 10.1038/s41565-021-00922-3). We suggest that the authors may elaborate on these methods in the introduction part and further highlight the advantages of afterglow imaging in this work.

Author reply: Thanks for the comments. According to the reviewer's suggestion, we have supplemented and elaborated on these imaging strategies (chemiluminescence, bioluminescence and afterglow), and further highlight the advantages of afterglow imaging in the revised manuscript. Please see “Currently, molecular imaging strategies such as bioluminescence,⁴⁸ chemiluminescence,⁴⁹ and afterglow luminescence can eliminate the requirement for spontaneous light irradiation, which have attracted the increasing interest. Most of bioluminescent probes are enzyme-dependent and chemiluminescent probes are based on flash-type Schaap's dioxetanes or Luminols.^{50, 51} And those bioluminescent or chemiluminescent probes showed the uncontrollable release of photons, which make it difficult to accurately detect biotargets in highly heterogeneous or dynamic biological scenarios. Interestingly, afterglow luminescence is externally controllable release of photons due to the separation of laser irradiation and photons acquisition, which holds great promise for sensitive and noninvasive imaging of biomolecules in living subjects.^{13, 22,}” in the revised discussion section.

2. The ARET-based ratiometric nanoplatfrom proposed by authors is composed of responsive molecules (NRM, ORM or PRM), afterglow substrate (MEHPPV),

surfactants (F127) and afterglow initiators (TPP or BDP) via a self-assembly strategy. What we are concerned about is the responsiveness difference between of ARET-based ratiometric nanoplatform and responsive molecules (NRM, ORM or PRM).

Author reply: According to the reviewer’s suggestion, we have supplemented the fluorescence imaging of responsive molecules (NRM, ORM or PRM) within different analyte concentrations, respectively. As shown in revised **Supplementary Figure 19**, there was no significant responsiveness difference between RAN1-2 and responsive molecules (NRM, ORM), through comparing the signal enhancement at maximum fluorescence emission channel. However, a significant responsiveness difference between RAN3 and PRM was observed, which may be ascribed to the larger overlap of emission spectral between PRM and MEHPPV in RAN3 system than that in RAN1,2 system (**Fig. 1d, f, h**). Please see “The spectra changes of absorption and fluorescence emission suggested the good responsiveness of RANs toward NO, ONOO⁻ and pH, respectively (Supplementary Fig. 18), and the responsive difference between ratiometric afterglow nanoplatform and responsive molecules (NRM, ORM or PRM) demonstrated that the selection of responsive molecules with larger wavelength gap with MEHPPV can reduce the spectral overlap between them, which is conducive to design ratiometric afterglow probes with better responsiveness (Supplementary Fig. 19).” in the revised manuscript.

Revised Supplementary Fig. 19. Fluorescence images of **a**, F127@NRM, **b**,

F127-PSMA@ORM, and **c**, F127@PRM in NO, ONOO⁻ buffer solutions with different concentrations and PBS with different pH values, respectively. **d**, The responsiveness difference between of ratiometric nanoplatform and responsive molecules (NRM, ORM or PRM) through comparing the signal enhancement at maximum fluorescence emission channel.

3. The imaging results illustrated in Figure 3k are different from the data in Figure 3n. As we all know, SBR compares the level of a desired signal to the level of background noise. In Figure 3n, the SBRs of AF2 signal under 0.1 cm chicken tissue and the AF1 signal under 0.6 cm chicken tissue are higher than that of FL2 under 0 cm chicken tissue. Hence, the signal should also be easily distinguished from the background in imaging results (Figure 3k). Here, we strongly recommend that the authors to zero the lower limit of the scalebar in Figure 3k and Supplementary Figure 21a to fully demonstrate the results of fluorescence imaging and make the data more consistent.

Author reply: According to the reviewer's suggestion, in the repeated experiment of the revised manuscript, we have reduced the lower limit of the scale bar, and a more consistent data and easily distinguishable imaging results were obtained from revised **Fig. 3k** and **Fig. 3n**, **Supplementary Fig. 23a** and **Supplementary Fig. 23d**.

Revised Fig. 3 k, Fluorescent and afterglow images of RAN1 (20 μg/mL) through chicken tissues of different thickness in the absence of NO. **n**, SBR for FL1, FL2, AF1, and AF2, as function of penetration depth in the absence of NO, respectively.

Revised Supplementary Fig. 23 a, Fluorescent and afterglow images of RAN1 (20 $\mu\text{g}/\text{mL}$) through chicken tissues of different thickness in the presence of NO (50 μM). **d**, SBR for FL1, FL2, AF1, and AF2 as function of penetration depth in (a), respectively.

4. The benchmarks for data normalization in Figures 3I and Supplementary Figure 21b are not clarified.

Author reply: Thanks for the comments. We defined the optical signal intensity as one unit with the chicken thickness at 0 cm and reprocessed **Fig. 3i** and **Supplementary Fig. 23b** in the revised manuscript to clearly clarify the imaging results.

Revised Fig. 3I, Normalized fluorescence intensities (FL1 and FL2) and afterglow luminescence intensities (AF1 and AF2), as a function of penetration depth in the

absence of NO. The signal intensity of chicken tissue with the thickness at 0 cm was defined as one unit.

Revised Supplementary Fig. 23 b, Normalized fluorescence intensities (FL1 and FL2) and afterglow luminescence intensities (AF1 and AF2) as a function of penetration depth in (a). The signal intensity of chicken tissue with the thickness at 0 cm was defined as one unit.

5. The results in Figure 3m are puzzling. Generally speaking, the photons with longer wavelength have deeper penetration depths in bio-tissues. However, FL2/FL1 in Figure show a downward trend, which means that the fluorescence emission intensity at 830 nm is more significantly affected by the depth of bio-tissue than the fluorescence emission at 600 nm. Authors should give a reasonable explanation for this phenomenon. On the other hand, the reason why the afterglow intensity ratio (AF2/AF1) remains constant at any depth from 0 to 0.6 cm also needs to be emphatically described. In our cognition, the attenuation rate of light intensity in a medium is basically only related to its wavelength. In Figure 3m, the significant difference between the fluorescence intensity ratio and the afterglow intensity is beyond our understanding. The authors should make it clear through further experiments or simulations.

Author reply: Thanks for the comments. As we all know, the acquired fluorescence signal is the sum of the fluorescence signal of the probe itself and the autofluorescence of chicken tissues, under laser irradiation. Obviously, the fluorescence of FL1 (600 nm) is higher than that of FL2 (830 nm) for chicken tissues, due to the higher autofluorescence under shorter wavelength for tissues. Thereby, in the repeated experiment of the revised manuscript, as the thickness of chicken tissues increasing, the autofluorescence intensity at FL1 (600 nm) would be significantly higher than that at FL2 (830 nm) for chicken tissue itself. As for the fluorescence signal of the probe itself, the ratio of FL2/FL1 would be no obviously changed, with the increasing the thickness of chicken tissues. Due to the addition of

autofluorescence of chicken tissues to the fluorescence of probe itself, the variation of total fluorescence signal of FL1 (autofluorescence FL1 + FL1 of probe) was larger than that of total FL2 (autofluorescence FL2 + FL2 of probe) (revised **Fig. 3i** and **Supplementary Fig. 23b**), which results in the attenuation of FL2/FL1 with the increase of thickness of chicken tissues.

In contrast, as for afterglow imaging, the auto-luminescence of chicken tissues is almost negligible, and the acquired afterglow signal are closer to the probe's signal, so the afterglow ratio (AF2/AF1) is less affected by the thickness of the chicken tissues. Therefore, we believe that the main reason for the difference in fluorescence and afterglow ratio may be attributed to the background signal of chicken tissues, which has been emphatically described in the revised manuscript.

6. In Figure 4b, why the AF2 signal in mouse injected with PBS stronger than that in mouse injected with LPS?

Author reply: Thanks for the comments. In this experiment, mice were intradermal (i.d.) injected with LPS into the dorsal skin of the rear paw. Due to the individual differences of mice, the enrichment ability and doses of the probe in the rear paws may be different for various mice. Because the single wavelength of afterglow signal was dependent on the probe's concentration, AF2 signal in mouse injected with PBS was significantly different with that in mouse injected with LPS. Even though, the ratiometric afterglow imaging strategy can effectively overcome the interference of probe's concentration. As a result, we can distinguish the inflammation group from the normal group through the change of the afterglow intensity ratio (AF2/AF1).

7. Supplementary figure 23a does not show the AF1 and FL1 signals properly after the mouse is injected with APAP.

Author reply: Thanks for the comments. We reprocessed the *in vivo* images of mice by using an independent color bar to show the AF1 and FL1 signals properly after the

mouse is injected with APAP. The imaging data was collected in the revised **Supplementary Figure 27a**.

Revised Supplementary Fig. 27. a, Representative images of BALB/c mice which received PBS and APAP (200 mg/kg, 100 μ L, intraperitoneally) for 1 h, followed by RAN2 (100 μ L, 40 μ g/mL, intravenously) for different durations, respectively. Scale bar: 2 cm.

8. Authors claimed that “RANI can monitor the fluctuations of intratumoral NO, as a biomarker of M1 macrophages”. Can this process be verified under a confocal microscope? It is recommended to verify the imaging advantages of RANI in other scenarios compared with ratio fluorescence probes.

Author reply: Thanks for the comments. It is reported that the intensity of afterglow luminescence was usually relatively low (0.1% - 1% of fluorescence), compared with that of fluorescence. Therefore, it is difficult to detect such low luminescence signal, using the commercial fluorescence confocal microscopy system, because no cold CCD camera were available for current microscopy system. Interestingly, the IVIS Lumina XR Imaging System with refrigerated CCD camera can indicate the weak

fluctuations of NO in macrophage polarization due to its high detection sensitivity. Thus, we have verified the fluctuation of NO during macrophage polarization in 96-well plates by Lumina XR Imaging System and obtained satisfactory imaging results, as shown in revised **Fig. 5b**.

In addition, according to the reviewer's suggestion, we have supplemented RAN1 for imaging of NO in LPS-induced liver injury model by using the *Nos2*^{-/-} mice. In the revised imaging experiment, mice were i.p. injected with LPS or PBS and then i.v. injected with RAN1. Then the mice were immediately imaged via an IVIS Lumina XR Imaging System. From the afterglow images and the quantification of liver areas, as for WT mice, the treatment of LPS could significantly enhance the afterglow intensity ratio (AF2/AF1), compared with PBS treatment, indicating the high content of NO in the liver of LPS-incubated WT mice group. As expected, as for *Nos2*^{-/-} mice, the treatment of LPS induced no obvious increase of afterglow intensity ratio (AF2/AF1), which is consistent with no notable production of NO. These results further confirming the feasibility of RAN1 for NO detection in liver injury through radiometric afterglow imaging (revised **Fig. 4d, e**).

Revised Fig. 4 d, Representative afterglow images of *Nos2*^{-/-} and WT mice upon post-intravenous injection of RAN1 (200 µg/mL) in the LPS-induced liver injury model. Scale bar: 2 cm. **e**, Normalized afterglow intensity ratio in (**d**).

Reviewer #2:

1. In the introduction, with references 7, 8 and 9 the authors should also add Liu et al, *Adv Drug Deliv Rev* 138 (2019) 193–210.

Author reply: According to the reviewer's suggestion, the reference was supplemented as **Reference 11** in the revised manuscript.

2. Do the authors have an idea of the amount of afterglow substrate, afterglow initiator, NRM, ORM or PRM incorporated into the nanoplatform? -Is this amount the same whatever the responsive molecule used? -What is the influence of the incorporated amount on the further ARET phenomenon?

Author reply: Thanks for the comments. We are able to measure the relative content of each component (afterglow substrate, afterglow initiator, and responsive molecules) in each nanoplatform by absorption spectra, using the typical absorption peak of each component. We ensured that each control group had the same absorbance in each assay.

Due to the different solubility of responsive molecules in organic solution, the responsive molecules in RAN system were different in each kind of nanoplatform. For example, compared with NRM and PRM, the weak hydrophobicity of ORM led to the relatively low loading content within RAN2 system.

As shown in **Fig. 3h, i**, at a certain concentration of NO, the afterglow ratio (AF2/AF1) was gradually increased with the doping amount of NRM increasing, which indicated the enhancement of ARET efficiency. Therefore, the ARET phenomenon was related with the doping amount of responsive molecule.

3. Fig S2 present the hydrodynamic diameter of RAN1. Is the size of RAN dependent on the responsive molecule incorporated into the nanoplatform? -In which medium was measured the hydrodynamic diameter? -Is the nanoplatform stable over the time? -Is the ARET phenomenon sensitive to the medium?

Author reply: Thanks for the comments. As shown in revised **Supplementary Fig.**

17c, there is no significant difference in the hydrodynamic diameter of RAN1, RAN2 and RAN3, indicating that the size of RAN is independent on the responsive molecule incorporated into the nanoplatform.

Revised Supplementary Fig. 11 b, Hydrodynamic diameter of RAN1 in DPBS, and cell culture medium during 5 days.

Revised Supplementary Fig. 17 c, The hydrodynamic diameter of RAN1, RAN2 and RAN3 in PBS buffer solutions.

The hydrodynamic diameter was measured in PBS buffer solutions. The stability of the nanoplatform was further explored through measuring the hydrodynamic diameter. As shown in revised **Supplementary Fig. 11b**, RAN exhibited good dispersion in DPBS and cell culture medium with diameters of ~45 nm during 5 days of measurement, indicating its good colloid stability.

We also tested the responsiveness of the nanoprobe (RAN1) in different medium (revised **Supplementary Fig. 7**), and there was no significant difference in the ratio of the afterglow intensity (AF2/AF1), indicating the ARET phenomenon is not sensitive to the medium.

Reviewer #3:

1. The title is non-specific and should be revised to communicate the main message

without jargon.

Author reply: According to the reviewer's suggestions, to communicate the main message of the manuscript, we have revised the title into "Ratiometric afterglow luminescent nanoplatform enables reliable quantification and molecular imaging" through removing relevant jargon, such as afterglow resonance energy transfer.

2. The abstract is not well written. For example, a term such as "afterglow" must be defined as this is non-standard. The authors' approach is also vague in many parts of the abstract. Terms such as "M1" should be avoided here as a general reader of "imaging" technology will have no idea about this.

Author reply: According to the reviewer's suggestions, to enhance the readability of this manuscript, we have re-written the abstract of the manuscript, including the defined term such as "afterglow", removed the term such as "M1", and some vague parts have also been revised.

3. In my view, the entire manuscript requires re-writing. It is not clear who the audience is, and the authors succumb to excessive jargon throughout. The manuscript is way too long for the content.

Author reply: Thanks for the comments. According the reviewer's suggestion, we have re-written the entire manuscript, made certain abridgements and necessary explanations for the jargon appearing in the manuscript, and simplified the content of the full text. In addition, in order to clearly show the meaning of jargon, we have summarized them in revised Supplementary Table S1. And, we believe the viewpoints and related experiments that we have presented are easily accessible to the audience in the field of molecular probes, biosensing and bioimaging, and cancer theranostic.

Supplementary Table S1 Explanation of the jargon in this manuscript.

Jargon	Explanation
--------	-------------

Afterglow	An internal luminescence pathway that occurs after photo-excitation
Afterglow substrate (MEHPPV)	Materials capable of generating afterglow luminescence after photo-excitation
Afterglow initiators (TPP or BDP)	Materials capable of inducing afterglow substrates to emit luminescence after photo-excitation
Responsive molecules (NRM, ORM, and PRM)	Activatable molecular probes that indicate the changes of targets (NO, ONOO ⁻ and pH)
Afterglow resonance energy transfer	A resonance energy transfer process between the afterglow substrate (energy donor) and the acceptor fluorophore (responsive molecule)
Ratiometric afterglow imaging	An afterglow imaging mode in which the two afterglow outputs exhibit simultaneously changing signals (eg. “seesaw” type) upon interaction with the targets
Macrophage polarization	Macrophages enable to change their activation states in response to growth factors and external cues, and potentially any other entity capable of being recognized by macrophages
Tumor microenvironment (TME)	The internal environment in which tumor cells generate and survive, and is the location of tumor occurrence, growth and metastasis
Tumor-associated macrophages (TAMs)	Macrophages that infiltrate in tumor tissue and are also the most abundant immune cell in the tumor microenvironment.
CD86, CD80, iNOS	Markers of M1-like macrophage

4. Similarly, I found the figures uninformative and requiring extensive revision. Take Figure 1a as an example: it is unclear what the authors seek to communicate in the left two boxes. It is also unnecessary to introduce “macrophage-mediated immunotherapy” in this figure (which the authors are not really doing, and this term is sufficiently nebulous to create confusion - figure 1c right box is simply a fantasy of the authors). Many of the figures are far, far too small and confusing.

Author reply: Thanks for the comments. We appropriately modified **Fig. 1a** and **1c** so that it can clearly show the construction of the ratiometric afterglow imaging nanoplatfrom and macrophage polarization process, respectively. To accurately show the application scenario of our ratiometric afterglow imaging nanoplatfrom, we have corrected “macrophage-mediated immunotherapy” into “macrophage polarization”. And according to the reviewer's suggestion, we have made appropriately enlargement

of the small figures.

Revised Fig. 1 Design of ratiometric luminescent nanoplatform for reliable imaging and evaluation of macrophage polarization. **a**, Schematic illustration of ARET-based ratiometric nanoplatform. **b**, Chemical structures of the responsive molecules (NRM, ORM and PRM) before and after response to NO for NRM, or ONOO⁻ for ORM, or pH for PRM. **c**, Real-time afterglow imaging of macrophage polarization.

5. When we talk about M1 macrophages in this case, the authors would be on safer ground if they defined what they were doing in the first place vis-à-vis stimulations. While it is true that iNOS is a “marker” of an M1-like macrophage, a better approach is to state this, and then continue with the actual stimulations. For example, define that M1-like macrophages express iNOS and then proceed by defining the actual stimulation conditions. Examples of the problems with nomenclature of macrophage

polarization have been published before (e.g. Murray et al. *Immunity*).

Author reply: Thanks for the comments. We have supplemented the immunofluorescence staining of iNOS (one of the markers for M1-phenotype macrophage and a key enzyme for NO generation) in RAW264.7 cells upon incubation with different polarization modulators. In this experiment, RAW 264.7 cells were treated with the different polarization modulators, including IFN- γ , BLZ945, pexidartinib and chloroquine. Then, the cells were fixed in 4% paraformaldehyde, permeabilized with 0.2% Triton X-100, blocked for nonspecific binding with 5% BSA in DPBS, followed by incubating with Anti-iNOS antibody. After washing and staining with secondary antibody, nucleus was stained with Hoechst 33258. Then the figures were analyzed by Olympus FV1000 laser confocal microscopy. The results showed that the expression level of iNOS in RAW264.7 cells incubated with IFN- γ was significantly higher than other polarization modulators (revised **Supplementary Fig. 32**), which is consistent with the NO level quantified by ratiometric afterglow imaging and level of M1-phenotype macrophage markers (CD86 and CD80) determined by flow data. And the nomenclature of macrophage polarization has been corrected in the revised manuscript according to the reviewer's suggested reference.

Revised Supplementary Fig. 32. Immunofluorescence staining of RAW264.7 cells

incubated with different polarization modulators. Green signals indicate iNOS stained with Anti-iNOS antibody and blue signals represent the cell nucleus (Hoechst 33258). Blue channel: $\lambda_{\text{ex}} = 405 \text{ nm}$, $\lambda_{\text{em}} = 425\text{-}475 \text{ nm}$; Green channel: $\lambda_{\text{ex}} = 488 \text{ nm}$, $\lambda_{\text{em}} = 500\text{-}560 \text{ nm}$. Scale bar: 20 μm .

6. While it seems that the RAN1 approach detects NO (and peroxynitrates), the authors can't safely conclude this with the RAW cell line. Instead, at least one in vivo experiment must be done using the *Nos2*^{-/-} mice compared to WT. This would include the flow data (including staining for intracellular iNOS).

Author reply: Thanks for the suggestion. We have supplemented RAN1 for imaging of NO in a lipopolysaccharide (LPS)-induced liver injury model by using the *Nos2*^{-/-} mice. In the imaging experiment, the *Nos2*^{-/-} mice and WT mice were divided into two groups (PBS group and LPS group). Mice were i.p. injected with LPS or PBS for 6 h and then i.v. injected with RAN1 for 60 min. The mice were immediately imaged via an IVIS Lumina XR Imaging System. For afterglow imaging, the mice were pre-illuminated with a 660-laser for 60 s. Then the laser was removed, and the afterglow images were acquired on an IVIS Spectrum imaging system with an acquisition time of 60 s. From the afterglow images and the quantification of liver areas, as for WT mice, the treatment of LPS could significantly enhance the afterglow intensity ratio (AF2/AF1), compared with PBS treatment, indicating the high content of NO in the liver of LPS-incubated WT mice group (revised **Fig. 4d, e**). As expected, as for *Nos2*^{-/-} mice, the treatment of LPS induced no obvious increase of afterglow intensity ratio (AF2/AF1), which is consistent with no notable production of NO.

For flow cytometric analysis and immunofluorescence staining of iNOS, the *Nos2*^{-/-} and WT mice were firstly i.p. injected with LPS or PBS for 6 h. Subsequently, the mice were sacrificed, and the liver were perfused with 0.05% collagenase IV through hepatic portal vein, and the livers were harvested after perfusion. Part of isolated liver tissue slices were acquired for immunofluorescence staining, using an Anti-iNOS antibody. And part of liver is minced and incubated with 0.05%

collagenase IV, and filtered with a 70 μm filter to obtain the single cell suspension. The cell suspension was centrifuged to obtain nonparenchymal cells, which were fixed in 4% paraformaldehyde, permeabilized with 0.2% Triton X-100, and blocked with 5% BSA, followed by incubating with Anti-iNOS antibody and secondary antibody. Finally, the samples were washed twice and analyzed using flow cytometry. The flow data and immunofluorescence staining of intracellular iNOS showed the higher expression of iNOS in LPS-incubated WT mice, while no obvious expression of iNOS in LPS-incubated *Nos2*^{-/-} mice (revised **Fig. 4f**, **Supplementary Fig. 26**). From those comparison experiment, we concluded that the RAN1 was able to detect endogenous NO in living mice via ratiometric afterglow imaging.

Revised Fig. 4 d, Representative afterglow images of *Nos2*^{-/-} and WT mice upon post-intravenous injection of RAN1 (200 $\mu\text{g/mL}$) in the LPS-induced liver injury model. Scale bar: 2 cm. **e**, Normalized afterglow intensity ratio in **(d)**. **f**, Representative immunofluorescence staining of iNOS in liver slices of *Nos2*^{-/-} and WT mice upon different administrations. Green signals indicate iNOS stained with Alexa Fluor 488-labelled iNOS antibody and blue signals represent the cell nucleus. Scale bar: 20 μm .

iNOS in liver of *Nos2*^{-/-} and WT mice upon different administrations.

7. The parts on “macrophage-mediated immunotherapy” must be revised to “manipulation” or “modulation”.

Author reply: According to the reviewer’s suggestion, “macrophage-mediated immunotherapy” has been corrected into “macrophage-modulated immunotherapy” in the revised manuscript.

8. The authors would improve the manuscript by making the first figure about the chemistry of the system, in more depth.

Author reply: According to the reviewer’s suggestion, the first figure has been modified by introducing the chemical response of the responsive molecules in the revised **Fig. 1b**.

Revised Fig. 1 b, Chemical structures of the responsive molecules (NRM, ORM and PRM) before and after response to NO for NRM, or ONOO⁻ for ORM, or pH for PRM.

9. Line 298. NO is not secreted; it freely diffuses.

Author reply: According to the reviewer’s suggestion, the error has been corrected in the revised manuscript.

Reviewer #4:

1. The response mechanism of RAN2 is different from that of RAN1 and RAN3. It is an “on-off” probe, and the characteristic emission of MEHPPV at 600 nm in RAN2 is significantly lower than that of RAN1 and RAN3, is this due to the higher energy transfer efficiency between MEHPPV and ORM in RAN2, or there are other reasons?

Author reply: Thanks for the comments. As shown in **Fig. 2c-g**, the fluorescence intensity of MEHPPV in RAN2 may appear to be weak, but in fact, the characteristic emission of MEHPPV at 600 nm in RAN2 is not significantly lower than that of RAN1 and RAN3. And the intensity difference of MEHPPV at 600 nm in RAN2 may be ascribed to the high doping amount of TPP in RAN2 system.

2. In RAN1 characterization, the important Zeta potential is absent, please provide its Zeta potential.

Author reply: According to the reviewer’s suggestion, the Zeta potential of RAN1 was supplemented in the revised manuscript. Please see the revised **Supplementary Fig. 2c**.

Revised Supplementary Fig. 2c, Zeta potential of RAN1

3. RAN1 was prepared by self-assembly, will it be disassembled in vivo? What is the stability in water or biological condition? More experiments should be provided to

indicate the stability of RAN1 in biological conditions.

Author reply: According to the reviewer's suggestion, the stability of the nanoplatform was further explored through measuring the hydrodynamic diameter. As shown in revised **Supplementary Fig. 11b**, RAN1 exhibited good dispersion in DPBS, and cell culture medium with diameters of ~45 nm during 5 days of measurement, indicating its good colloid stability.

Revised Supplementary Fig. 11 b, Hydrodynamic diameter of RAN1 in DPBS, and cell culture medium during 5 days.

4. What is the purpose of introducing these AIs? What effect does it have on the ARET process?

Author reply: Thanks for the comments. Afterglow initiators (AIs) are also photosensitizers, and they are able to induce afterglow substrates to emit luminescence through generating singlet oxygen. In addition, they can not only extend the pre-irradiation wavelength of the ratiometric afterglow nanoprobe, but also serve as an energy transfer bridge between the afterglow substrate and the responsive molecules to improve the ARET efficiency between them.

5. The UV-visible absorption and fluorescence emission spectra of NRM, ORM and PRM before and after the response should be provided.

Author reply: According to the reviewer's suggestion, the UV-visible absorption and fluorescence emission spectra of NRM, ORM and PRM before and after the response were supplemented in the revised manuscript. Please see the revised **Supplementary Fig. 18**.

Revised Supplementary Fig. 18. a-c, d-f, Absorption (a-c) and fluorescence spectra (d-f) of 5 μM NRM, ORM, and PRM in the absence or presence of their respective targets (25 μM NO, 4.0 μM ONOO⁻ and pH 3.0, respectively) in PBS solutions. The fluorescence excitation was set at 660 nm, 760 nm, and 710 nm, respectively.

6. In Fig. 4b, the afterglow signals of the rear paw are not clearly displayed. I recommend the authors to enlarge the signal area to obtain high-quality imaging pictures.

Author reply: According to the reviewer's suggestion, the signal area of the rear paw was enlarged and high-quality imaging pictures was obtained in the revised Fig. 4b.

7. Supplementary Fig. 21d showed that the SBR of AF1 was similar to that of AF2 in *in vitro* experiment, however, the SBR of AF2 was significantly lower than that of AF1 in *in vivo* experiment (Fig. 4h). What caused the difference between *in vivo* and *in*

vitro experiments?

Author reply: Thanks for the comments. The magnitude of the SBR ratio is determined by the intensity of the signal and the background, in which the signal intensity is mainly dependent on the concentration of the analyte (NO), and the background intensity mainly depends on the auto-luminescence of the biological tissue. In afterglow imaging, the auto-luminescence of chicken tissues (in vitro) and mice tumor (in vivo) is almost negligible. Therefore, the concentration of NO determined the difference between in vivo and in vitro experiments. In the experiment of revised **Supplementary Fig. 23d**, RAN1 solutions were placed under chicken tissues of varying thickness, and the high NO concentration can induce similar intensity of AF1 and AF2, thereby result in similar SBR. While in the experiment of revised **Fig. 4h**, RAN1 was enriched in tumors through i.v. injection, the intensity and SBR of AF2 is significantly weaker than that of AF1, due to the low endogenous content of NO in tumors. Therefore, the difference of SBR between in vivo and in vitro experiments may be mainly ascribed to the difference in NO content.

8. To show the size of the mice, I recommend the authors to add a scale bar to the appropriate position on all the imaging pictures of mice, including Fig. 4b, d, g, Fig. 5e, Supplementary Fig. 22a, 23a, 28a.

Author reply: According to the reviewer's suggestion, the scale bar was supplemented on all the imaging pictures of mice. Please see revised **Fig. 4b, d, g, i Fig. 5e, Supplementary Fig. 24a, 25a, 27a, 33a.**

9. The format of the references needs to be checked carefully, and some unnecessary errors should be avoided as much as possible, such as "tumor-to-Liver" in Ref. 16, "real-Time" in Ref. 40.

Author reply: According to the reviewer's suggestion, the errors of the references were corrected in the revised manuscript.

Reviewers' Comments:

Reviewer #1:

Remarks to the Author:

The authors have revised the manuscript properly based on the reviewer's comments. I recommend to accept it in the present form.

Reviewer #2:

Remarks to the Author:

The authors answered all of the reviewer's questions.

Reviewer #3:

Remarks to the Author:

The authors have addressed all the main points raised (and improved the figures)

Reviewer #4:

Remarks to the Author:

After nearly half a year, the authors have well addressed the comments raised, including the Nos2-/- mice. The paper is improved over previous one.

Point-by-point responses to the reviewers' comments:

Reviewer #1:

1. The shortages of traditional fluorescence imaging are introduced in the introduction. However, recent works have indicated that there are some optical imaging strategies that can overcome these shortcomings, such as chemiluminescence imaging (e.g. 10.1002/anie.202007649), bioluminescence imaging (e.g. 10.1038/s41467-020-18051-1), and afterglow imaging (e.g. 10.1038/s41565-021-00922-3). We suggest that the authors may elaborate on these methods in the introduction part and further highlight the advantages of afterglow imaging in this work.

Author reply: Thanks for the comments. According to the reviewer's suggestion, we have supplemented and elaborated on these imaging strategies (chemiluminescence, bioluminescence and afterglow), and further highlight the advantages of afterglow imaging in the revised manuscript. Please see “Currently, molecular imaging strategies such as bioluminescence,⁴⁸ chemiluminescence,⁴⁹ and afterglow luminescence can eliminate the requirement for spontaneous light irradiation, which have attracted the increasing interest. Most of bioluminescent probes are enzyme-dependent and chemiluminescent probes are based on flash-type Schaap's dioxetanes or Luminols.^{50, 51} And those bioluminescent or chemiluminescent probes showed the uncontrollable release of photons, which make it difficult to accurately detect biotargets in highly heterogeneous or dynamic biological scenarios. Interestingly, afterglow luminescence is externally controllable release of photons due to the separation of laser irradiation and photons acquisition, which holds great promise for sensitive and noninvasive imaging of biomolecules in living subjects.^{13, 22,}” in the revised discussion section.

2. The ARET-based ratiometric nanoplatfrom proposed by authors is composed of responsive molecules (NRM, ORM or PRM), afterglow substrate (MEHPPV),

surfactants (F127) and afterglow initiators (TPP or BDP) via a self-assembly strategy. What we are concerned about is the responsiveness difference between of ARET-based ratiometric nanoplatform and responsive molecules (NRM, ORM or PRM).

Author reply: According to the reviewer’s suggestion, we have supplemented the fluorescence imaging of responsive molecules (NRM, ORM or PRM) within different analyte concentrations, respectively. As shown in revised **Supplementary Figure 19**, there was no significant responsiveness difference between RAN1-2 and responsive molecules (NRM, ORM), through comparing the signal enhancement at maximum fluorescence emission channel. However, a significant responsiveness difference between RAN3 and PRM was observed, which may be ascribed to the larger overlap of emission spectral between PRM and MEHPPV in RAN3 system than that in RAN1,2 system (**Fig. 1d, f, h**). Please see “The spectra changes of absorption and fluorescence emission suggested the good responsiveness of RANs toward NO, ONOO- and pH, respectively (Supplementary Fig. 18), and the responsive difference between ratiometric afterglow nanoplatform and responsive molecules (NRM, ORM or PRM) demonstrated that the selection of responsive molecules with larger wavelength gap with MEHPPV can reduce the spectral overlap between them, which is conducive to design ratiometric afterglow probes with better responsiveness (Supplementary Fig. 19).” in the revised manuscript.

Revised Supplementary Fig. 19. Fluorescence images of **a**, F127@NRM, **b**,

F127-PSMA@ORM, and **c**, F127@PRM in NO, ONOO⁻ buffer solutions with different concentrations and PBS with different pH values, respectively. **d**, The responsiveness difference between of ratiometric nanoplatform and responsive molecules (NRM, ORM or PRM) through comparing the signal enhancement at maximum fluorescence emission channel.

3. The imaging results illustrated in Figure 3k are different from the data in Figure 3n. As we all know, SBR compares the level of a desired signal to the level of background noise. In Figure 3n, the SBRs of AF2 signal under 0.1 cm chicken tissue and the AF1 signal under 0.6 cm chicken tissue are higher than that of FL2 under 0 cm chicken tissue. Hence, the signal should also be easily distinguished from the background in imaging results (Figure 3k). Here, we strongly recommend that the authors to zero the lower limit of the scalebar in Figure 3k and Supplementary Figure 21a to fully demonstrate the results of fluorescence imaging and make the data more consistent.

Author reply: According to the reviewer's suggestion, in the repeated experiment of the revised manuscript, we have reduced the lower limit of the scale bar, and a more consistent data and easily distinguishable imaging results were obtained from revised **Fig. 3k** and **Fig. 3n**, **Supplementary Fig. 23a** and **Supplementary Fig. 23d**.

Revised Fig. 3 k, Fluorescent and afterglow images of RAN1 (20 μg/mL) through chicken tissues of different thickness in the absence of NO. **n**, SBR for FL1, FL2, AF1, and AF2, as function of penetration depth in the absence of NO, respectively.

Revised Supplementary Fig. 23 a, Fluorescent and afterglow images of RAN1 (20 $\mu\text{g/mL}$) through chicken tissues of different thickness in the presence of NO (50 μM). **d**, SBR for FL1, FL2, AF1, and AF2 as function of penetration depth in (a), respectively.

4. The benchmarks for data normalization in Figures 3I and Supplementary Figure 21b are not clarified.

Author reply: Thanks for the comments. We defined the optical signal intensity as one unit with the chicken thickness at 0 cm and reprocessed **Fig. 3i** and **Supplementary Fig. 23b** in the revised manuscript to clearly clarify the imaging results.

Revised Fig. 3I, Normalized fluorescence intensities (FL1 and FL2) and afterglow luminescence intensities (AF1 and AF2), as a function of penetration depth in the

absence of NO. The signal intensity of chicken tissue with the thickness at 0 cm was defined as one unit.

Revised Supplementary Fig. 23 b, Normalized fluorescence intensities (FL1 and FL2) and afterglow luminescence intensities (AF1 and AF2) as a function of penetration depth in (a). The signal intensity of chicken tissue with the thickness at 0 cm was defined as one unit.

5. The results in Figure 3m are puzzling. Generally speaking, the photons with longer wavelength have deeper penetration depths in bio-tissues. However, FL2/FL1 in Figure show a downward trend, which means that the fluorescence emission intensity at 830 nm is more significantly affected by the depth of bio-tissue than the fluorescence emission at 600 nm. Authors should give a reasonable explanation for this phenomenon. On the other hand, the reason why the afterglow intensity ratio (AF2/AF1) remains constant at any depth from 0 to 0.6 cm also needs to be emphatically described. In our cognition, the attenuation rate of light intensity in a medium is basically only related to its wavelength. In Figure 3m, the significant difference between the fluorescence intensity ratio and the afterglow intensity is beyond our understanding. The authors should make it clear through further experiments or simulations.

Author reply: Thanks for the comments. As we all know, the acquired fluorescence signal is the sum of the fluorescence signal of the probe itself and the autofluorescence of chicken tissues, under laser irradiation. Obviously, the fluorescence of FL1 (600 nm) is higher than that of FL2 (830 nm) for chicken tissues, due to the higher autofluorescence under shorter wavelength for tissues. Thereby, in the repeated experiment of the revised manuscript, as the thickness of chicken tissues increasing, the autofluorescence intensity at FL1 (600 nm) would be significantly higher than that at FL2 (830 nm) for chicken tissue itself. As for the fluorescence signal of the probe itself, the ratio of FL2/FL1 would be no obviously changed, with the increasing the thickness of chicken tissues. Due to the addition of

autofluorescence of chicken tissues to the fluorescence of probe itself, the variation of total fluorescence signal of FL1 (autofluorescence FL1 + FL1 of probe) was larger than that of total FL2 (autofluorescence FL2 + FL2 of probe) (revised **Fig. 3i** and **Supplementary Fig. 23b**), which results in the attenuation of FL2/FL1 with the increase of thickness of chicken tissues.

In contrast, as for afterglow imaging, the auto-luminescence of chicken tissues is almost negligible, and the acquired afterglow signal are closer to the probe's signal, so the afterglow ratio (AF2/AF1) is less affected by the thickness of the chicken tissues. Therefore, we believe that the main reason for the difference in fluorescence and afterglow ratio may be attributed to the background signal of chicken tissues, which has been emphatically described in the revised manuscript.

6. In Figure 4b, why the AF2 signal in mouse injected with PBS stronger than that in mouse injected with LPS?

Author reply: Thanks for the comments. In this experiment, mice were intradermal (i.d.) injected with LPS into the dorsal skin of the rear paw. Due to the individual differences of mice, the enrichment ability and doses of the probe in the rear paws may be different for various mice. Because the single wavelength of afterglow signal was dependent on the probe's concentration, AF2 signal in mouse injected with PBS was significantly different with that in mouse injected with LPS. Even though, the ratiometric afterglow imaging strategy can effectively overcome the interference of probe's concentration. As a result, we can distinguish the inflammation group from the normal group through the change of the afterglow intensity ratio (AF2/AF1).

7. Supplementary figure 23a does not show the AF1 and FL1 signals properly after the mouse is injected with APAP.

Author reply: Thanks for the comments. We reprocessed the *in vivo* images of mice by using an independent color bar to show the AF1 and FL1 signals properly after the

mouse is injected with APAP. The imaging data was collected in the revised **Supplementary Figure 27a**.

Revised Supplementary Fig. 27. a, Representative images of BALB/c mice which received PBS and APAP (200 mg/kg, 100 μ L, intraperitoneally) for 1 h, followed by RAN2 (100 μ L, 40 μ g/mL, intravenously) for different durations, respectively. Scale bar: 2 cm.

8. Authors claimed that “RANI can monitor the fluctuations of intratumoral NO, as a biomarker of M1 macrophages”. Can this process be verified under a confocal microscope? It is recommended to verify the imaging advantages of RANI in other scenarios compared with ratio fluorescence probes.

Author reply: Thanks for the comments. It is reported that the intensity of afterglow luminescence was usually relatively low (0.1% - 1% of fluorescence), compared with that of fluorescence. Therefore, it is difficult to detect such low luminescence signal, using the commercial fluorescence confocal microscopy system, because no cold CCD camera were available for current microscopy system. Interestingly, the IVIS Lumina XR Imaging System with refrigerated CCD camera can indicate the weak

fluctuations of NO in macrophage polarization due to its high detection sensitivity. Thus, we have verified the fluctuation of NO during macrophage polarization in 96-well plates by Lumina XR Imaging System and obtained satisfactory imaging results, as shown in revised **Fig. 5b**.

In addition, according to the reviewer's suggestion, we have supplemented RAN1 for imaging of NO in LPS-induced liver injury model by using the *Nos2*^{-/-} mice. In the revised imaging experiment, mice were i.p. injected with LPS or PBS and then i.v. injected with RAN1. Then the mice were immediately imaged via an IVIS Lumina XR Imaging System. From the afterglow images and the quantification of liver areas, as for WT mice, the treatment of LPS could significantly enhance the afterglow intensity ratio (AF2/AF1), compared with PBS treatment, indicating the high content of NO in the liver of LPS-incubated WT mice group. As expected, as for *Nos2*^{-/-} mice, the treatment of LPS induced no obvious increase of afterglow intensity ratio (AF2/AF1), which is consistent with no notable production of NO. These results further confirming the feasibility of RAN1 for NO detection in liver injury through radiometric afterglow imaging (revised **Fig. 4d, e**).

Revised Fig. 4 d, Representative afterglow images of *Nos2*^{-/-} and WT mice upon post-intravenous injection of RAN1 (200 µg/mL) in the LPS-induced liver injury model. Scale bar: 2 cm. **e**, Normalized afterglow intensity ratio in (**d**).

Reviewer #2:

1. In the introduction, with references 7, 8 and 9 the authors should also add Liu et al, *Adv Drug Deliv Rev* 138 (2019) 193–210.

Author reply: According to the reviewer's suggestion, the reference was supplemented as **Reference 11** in the revised manuscript.

2. Do the authors have an idea of the amount of afterglow substrate, afterglow initiator, NRM, ORM or PRM incorporated into the nanoplatform? -Is this amount the same whatever the responsive molecule used? -What is the influence of the incorporated amount on the further ARET phenomenon?

Author reply: Thanks for the comments. We are able to measure the relative content of each component (afterglow substrate, afterglow initiator, and responsive molecules) in each nanoplatform by absorption spectra, using the typical absorption peak of each component. We ensured that each control group had the same absorbance in each assay.

Due to the different solubility of responsive molecules in organic solution, the responsive molecules in RAN system were different in each kind of nanoplatform. For example, compared with NRM and PRM, the weak hydrophobicity of ORM led to the relatively low loading content within RAN2 system.

As shown in **Fig. 3h, i**, at a certain concentration of NO, the afterglow ratio (AF2/AF1) was gradually increased with the doping amount of NRM increasing, which indicated the enhancement of ARET efficiency. Therefore, the ARET phenomenon was related with the doping amount of responsive molecule.

3. Fig S2 present the hydrodynamic diameter of RAN1. Is the size of RAN dependent on the responsive molecule incorporated into the nanoplatform? -In which medium was measured the hydrodynamic diameter? -Is the nanoplatform stable over the time? -Is the ARET phenomenon sensitive to the medium?

Author reply: Thanks for the comments. As shown in revised **Supplementary Fig.**

17c, there is no significant difference in the hydrodynamic diameter of RAN1, RAN2 and RAN3, indicating that the size of RAN is independent on the responsive molecule incorporated into the nanoplatform.

Revised Supplementary Fig. 11 b, Hydrodynamic diameter of RAN1 in DPBS, and cell culture medium during 5 days.

Revised Supplementary Fig. 17 c, The hydrodynamic diameter of RAN1, RAN2 and RAN3 in PBS buffer solutions.

The hydrodynamic diameter was measured in PBS buffer solutions. The stability of the nanoplatform was further explored through measuring the hydrodynamic diameter. As shown in revised **Supplementary Fig. 11b**, RAN exhibited good dispersion in DPBS and cell culture medium with diameters of ~45 nm during 5 days of measurement, indicating its good colloid stability.

We also tested the responsiveness of the nanoprobe (RAN1) in different medium (revised **Supplementary Fig. 7**), and there was no significant difference in the ratio of the afterglow intensity (AF2/AF1), indicating the ARET phenomenon is not sensitive to the medium.

Reviewer #3:

1. The title is non-specific and should be revised to communicate the main message

without jargon.

Author reply: According to the reviewer's suggestions, to communicate the main message of the manuscript, we have revised the title into "Ratiometric afterglow luminescent nanoplatform enables reliable quantification and molecular imaging" through removing relevant jargon, such as afterglow resonance energy transfer.

2. The abstract is not well written. For example, a term such as "afterglow" must be defined as this is non-standard. The authors' approach is also vague in many parts of the abstract. Terms such as "M1" should be avoided here as a general reader of "imaging" technology will have no idea about this.

Author reply: According to the reviewer's suggestions, to enhance the readability of this manuscript, we have re-written the abstract of the manuscript, including the defined term such as "afterglow", removed the term such as "M1", and some vague parts have also been revised.

3. In my view, the entire manuscript requires re-writing. It is not clear who the audience is, and the authors succumb to excessive jargon throughout. The manuscript is way too long for the content.

Author reply: Thanks for the comments. According the reviewer's suggestion, we have re-written the entire manuscript, made certain abridgements and necessary explanations for the jargon appearing in the manuscript, and simplified the content of the full text. In addition, in order to clearly show the meaning of jargon, we have summarized them in revised Supplementary Table S1. And, we believe the viewpoints and related experiments that we have presented are easily accessible to the audience in the field of molecular probes, biosensing and bioimaging, and cancer theranostic.

Supplementary Table S1 Explanation of the jargon in this manuscript.

Jargon	Explanation
--------	-------------

Afterglow	An internal luminescence pathway that occurs after photo-excitation
Afterglow substrate (MEHPPV)	Materials capable of generating afterglow luminescence after photo-excitation
Afterglow initiators (TPP or BDP)	Materials capable of inducing afterglow substrates to emit luminescence after photo-excitation
Responsive molecules (NRM, ORM, and PRM)	Activatable molecular probes that indicate the changes of targets (NO, ONOO ⁻ and pH)
Afterglow resonance energy transfer	A resonance energy transfer process between the afterglow substrate (energy donor) and the acceptor fluorophore (responsive molecule)
Ratiometric afterglow imaging	An afterglow imaging mode in which the two afterglow outputs exhibit simultaneously changing signals (eg. “seesaw” type) upon interaction with the targets
Macrophage polarization	Macrophages enable to change their activation states in response to growth factors and external cues, and potentially any other entity capable of being recognized by macrophages
Tumor microenvironment (TME)	The internal environment in which tumor cells generate and survive, and is the location of tumor occurrence, growth and metastasis
Tumor-associated macrophages (TAMs)	Macrophages that infiltrate in tumor tissue and are also the most abundant immune cell in the tumor microenvironment.
CD86, CD80, iNOS	Markers of M1-like macrophage

4. Similarly, I found the figures uninformative and requiring extensive revision. Take Figure 1a as an example: it is unclear what the authors seek to communicate in the left two boxes. It is also unnecessary to introduce “macrophage-mediated immunotherapy” in this figure (which the authors are not really doing, and this term is sufficiently nebulous to create confusion - figure 1c right box is simply a fantasy of the authors). Many of the figures are far, far too small and confusing.

Author reply: Thanks for the comments. We appropriately modified **Fig. 1a** and **1c** so that it can clearly show the construction of the ratiometric afterglow imaging nanoplatfrom and macrophage polarization process, respectively. To accurately show the application scenario of our ratiometric afterglow imaging nanoplatfrom, we have corrected “macrophage-mediated immunotherapy” into “macrophage polarization”. And according to the reviewer's suggestion, we have made appropriately enlargement

of the small figures.

Revised Fig. 1 Design of ratiometric luminescent nanoplatform for reliable imaging and evaluation of macrophage polarization. **a**, Schematic illustration of ARET-based ratiometric nanoplatform. **b**, Chemical structures of the responsive molecules (NRM, ORM and PRM) before and after response to NO for NRM, or ONOO⁻ for ORM, or pH for PRM. **c**, Real-time afterglow imaging of macrophage polarization.

5. When we talk about M1 macrophages in this case, the authors would be on safer ground if they defined what they were doing in the first place vis-à-vis stimulations. While it is true that iNOS is a “marker” of an M1-like macrophage, a better approach is to state this, and then continue with the actual stimulations. For example, define that M1-like macrophages express iNOS and then proceed by defining the actual stimulation conditions. Examples of the problems with nomenclature of macrophage

polarization have been published before (e.g. Murray et al. *Immunity*).

Author reply: Thanks for the comments. We have supplemented the immunofluorescence staining of iNOS (one of the markers for M1-phenotype macrophage and a key enzyme for NO generation) in RAW264.7 cells upon incubation with different polarization modulators. In this experiment, RAW 264.7 cells were treated with the different polarization modulators, including IFN- γ , BLZ945, pexidartinib and chloroquine. Then, the cells were fixed in 4% paraformaldehyde, permeabilized with 0.2% Triton X-100, blocked for nonspecific binding with 5% BSA in DPBS, followed by incubating with Anti-iNOS antibody. After washing and staining with secondary antibody, nucleus was stained with Hoechst 33258. Then the figures were analyzed by Olympus FV1000 laser confocal microscopy. The results showed that the expression level of iNOS in RAW264.7 cells incubated with IFN- γ was significantly higher than other polarization modulators (revised **Supplementary Fig. 32**), which is consistent with the NO level quantified by ratiometric afterglow imaging and level of M1-phenotype macrophage markers (CD86 and CD80) determined by flow data. And the nomenclature of macrophage polarization has been corrected in the revised manuscript according to the reviewer's suggested reference.

Revised Supplementary Fig. 32. Immunofluorescence staining of RAW264.7 cells

incubated with different polarization modulators. Green signals indicate iNOS stained with Anti-iNOS antibody and blue signals represent the cell nucleus (Hoechst 33258). Blue channel: $\lambda_{\text{ex}} = 405 \text{ nm}$, $\lambda_{\text{em}} = 425\text{-}475 \text{ nm}$; Green channel: $\lambda_{\text{ex}} = 488 \text{ nm}$, $\lambda_{\text{em}} = 500\text{-}560 \text{ nm}$. Scale bar: 20 μm .

6. While it seems that the RAN1 approach detects NO (and peroxynitrates), the authors can't safely conclude this with the RAW cell line. Instead, at least one in vivo experiment must be done using the *Nos2*^{-/-} mice compared to WT. This would include the flow data (including staining for intracellular iNOS).

Author reply: Thanks for the suggestion. We have supplemented RAN1 for imaging of NO in a lipopolysaccharide (LPS)-induced liver injury model by using the *Nos2*^{-/-} mice. In the imaging experiment, the *Nos2*^{-/-} mice and WT mice were divided into two groups (PBS group and LPS group). Mice were i.p. injected with LPS or PBS for 6 h and then i.v. injected with RAN1 for 60 min. The mice were immediately imaged via an IVIS Lumina XR Imaging System. For afterglow imaging, the mice were pre-illuminated with a 660-laser for 60 s. Then the laser was removed, and the afterglow images were acquired on an IVIS Spectrum imaging system with an acquisition time of 60 s. From the afterglow images and the quantification of liver areas, as for WT mice, the treatment of LPS could significantly enhance the afterglow intensity ratio (AF2/AF1), compared with PBS treatment, indicating the high content of NO in the liver of LPS-incubated WT mice group (revised **Fig. 4d, e**). As expected, as for *Nos2*^{-/-} mice, the treatment of LPS induced no obvious increase of afterglow intensity ratio (AF2/AF1), which is consistent with no notable production of NO.

For flow cytometric analysis and immunofluorescence staining of iNOS, the *Nos2*^{-/-} and WT mice were firstly i.p. injected with LPS or PBS for 6 h. Subsequently, the mice were sacrificed, and the liver were perfused with 0.05% collagenase IV through hepatic portal vein, and the livers were harvested after perfusion. Part of isolated liver tissue slices were acquired for immunofluorescence staining, using an Anti-iNOS antibody. And part of liver is minced and incubated with 0.05%

collagenase IV, and filtered with a 70 μm filter to obtain the single cell suspension. The cell suspension was centrifuged to obtain nonparenchymal cells, which were fixed in 4% paraformaldehyde, permeabilized with 0.2% Triton X-100, and blocked with 5% BSA, followed by incubating with Anti-iNOS antibody and secondary antibody. Finally, the samples were washed twice and analyzed using flow cytometry. The flow data and immunofluorescence staining of intracellular iNOS showed the higher expression of iNOS in LPS-incubated WT mice, while no obvious expression of iNOS in LPS-incubated *Nos2*^{-/-} mice (revised **Fig. 4f**, **Supplementary Fig. 26**). From those comparison experiment, we concluded that the RAN1 was able to detect endogenous NO in living mice via ratiometric afterglow imaging.

Revised Fig. 4 d, Representative afterglow images of *Nos2*^{-/-} and WT mice upon post-intravenous injection of RAN1 (200 $\mu\text{g/mL}$) in the LPS-induced liver injury model. Scale bar: 2 cm. **e**, Normalized afterglow intensity ratio in **(d)**. **f**, Representative immunofluorescence staining of iNOS in liver slices of *Nos2*^{-/-} and WT mice upon different administrations. Green signals indicate iNOS stained with Alexa Fluor 488-labelled iNOS antibody and blue signals represent the cell nucleus. Scale bar: 20 μm .

iNOS in liver of *Nos2*^{-/-} and WT mice upon different administrations.

7. The parts on “macrophage-mediated immunotherapy” must be revised to “manipulation” or “modulation”.

Author reply: According to the reviewer’s suggestion, “macrophage-mediated immunotherapy” has been corrected into “macrophage-modulated immunotherapy” in the revised manuscript.

8. The authors would improve the manuscript by making the first figure about the chemistry of the system, in more depth.

Author reply: According to the reviewer’s suggestion, the first figure has been modified by introducing the chemical response of the responsive molecules in the revised **Fig. 1b**.

Revised Fig. 1 b, Chemical structures of the responsive molecules (NRM, ORM and PRM) before and after response to NO for NRM, or ONOO⁻ for ORM, or pH for PRM.

9. Line 298. NO is not secreted; it freely diffuses.

Author reply: According to the reviewer’s suggestion, the error has been corrected in the revised manuscript.

Reviewer #4:

1. The response mechanism of RAN2 is different from that of RAN1 and RAN3. It is an “on-off” probe, and the characteristic emission of MEHPPV at 600 nm in RAN2 is significantly lower than that of RAN1 and RAN3, is this due to the higher energy transfer efficiency between MEHPPV and ORM in RAN2, or there are other reasons?

Author reply: Thanks for the comments. As shown in **Fig. 2c-g**, the fluorescence intensity of MEHPPV in RAN2 may appear to be weak, but in fact, the characteristic emission of MEHPPV at 600 nm in RAN2 is not significantly lower than that of RAN1 and RAN3. And the intensity difference of MEHPPV at 600 nm in RAN2 may be ascribed to the high doping amount of TPP in RAN2 system.

2. In RAN1 characterization, the important Zeta potential is absent, please provide its Zeta potential.

Author reply: According to the reviewer’s suggestion, the Zeta potential of RAN1 was supplemented in the revised manuscript. Please see the revised **Supplementary Fig. 2c**.

Revised Supplementary Fig. 2c, Zeta potential of RAN1

3. RAN1 was prepared by self-assembly, will it be disassembled in vivo? What is the stability in water or biological condition? More experiments should be provided to

indicate the stability of RAN1 in biological conditions.

Author reply: According to the reviewer's suggestion, the stability of the nanoplatform was further explored through measuring the hydrodynamic diameter. As shown in revised **Supplementary Fig. 11b**, RAN1 exhibited good dispersion in DPBS, and cell culture medium with diameters of ~45 nm during 5 days of measurement, indicating its good colloid stability.

Revised Supplementary Fig. 11 b, Hydrodynamic diameter of RAN1 in DPBS, and cell culture medium during 5 days.

4. What is the purpose of introducing these AIs? What effect does it have on the ARET process?

Author reply: Thanks for the comments. Afterglow initiators (AIs) are also photosensitizers, and they are able to induce afterglow substrates to emit luminescence through generating singlet oxygen. In addition, they can not only extend the pre-irradiation wavelength of the ratiometric afterglow nanoprobe, but also serve as an energy transfer bridge between the afterglow substrate and the responsive molecules to improve the ARET efficiency between them.

5. The UV-visible absorption and fluorescence emission spectra of NRM, ORM and PRM before and after the response should be provided.

Author reply: According to the reviewer's suggestion, the UV-visible absorption and fluorescence emission spectra of NRM, ORM and PRM before and after the response were supplemented in the revised manuscript. Please see the revised **Supplementary Fig. 18**.

Revised Supplementary Fig. 18. a-c, d-f, Absorption (a-c) and fluorescence spectra (d-f) of 5 μM NRM, ORM, and PRM in the absence or presence of their respective targets (25 μM NO, 4.0 μM ONOO⁻ and pH 3.0, respectively) in PBS solutions. The fluorescence excitation was set at 660 nm, 760 nm, and 710 nm, respectively.

6. In Fig. 4b, the afterglow signals of the rear paw are not clearly displayed. I recommend the authors to enlarge the signal area to obtain high-quality imaging pictures.

Author reply: According to the reviewer's suggestion, the signal area of the rear paw was enlarged and high-quality imaging pictures was obtained in the revised Fig. 4b.

7. Supplementary Fig. 21d showed that the SBR of AF1 was similar to that of AF2 in *in vitro* experiment, however, the SBR of AF2 was significantly lower than that of AF1 in *in vivo* experiment (Fig. 4h). What caused the difference between *in vivo* and *in*

vitro experiments?

Author reply: Thanks for the comments. The magnitude of the SBR ratio is determined by the intensity of the signal and the background, in which the signal intensity is mainly dependent on the concentration of the analyte (NO), and the background intensity mainly depends on the auto-luminescence of the biological tissue. In afterglow imaging, the auto-luminescence of chicken tissues (in vitro) and mice tumor (in vivo) is almost negligible. Therefore, the concentration of NO determined the difference between in vivo and in vitro experiments. In the experiment of revised **Supplementary Fig. 23d**, RAN1 solutions were placed under chicken tissues of varying thickness, and the high NO concentration can induce similar intensity of AF1 and AF2, thereby result in similar SBR. While in the experiment of revised **Fig. 4h**, RAN1 was enriched in tumors through i.v. injection, the intensity and SBR of AF2 is significantly weaker than that of AF1, due to the low endogenous content of NO in tumors. Therefore, the difference of SBR between in vivo and in vitro experiments may be mainly ascribed to the difference in NO content.

8. To show the size of the mice, I recommend the authors to add a scale bar to the appropriate position on all the imaging pictures of mice, including Fig. 4b, d, g, Fig. 5e, Supplementary Fig. 22a, 23a, 28a.

Author reply: According to the reviewer's suggestion, the scale bar was supplemented on all the imaging pictures of mice. Please see revised **Fig. 4b, d, g, i Fig. 5e, Supplementary Fig. 24a, 25a, 27a, 33a.**

9. The format of the references needs to be checked carefully, and some unnecessary errors should be avoided as much as possible, such as "tumor-to-Liver" in Ref. 16, "real-Time" in Ref. 40.

Author reply: According to the reviewer's suggestion, the errors of the references were corrected in the revised manuscript.